# PiCO: Contrastive Label Disambiguation for Partial Label Learning

**Haobo Wang**[1]   **Ruixuan Xiao**[1]   **Yixuan Li**[2]   **Lei Feng**[34]
**Gang Niu**[4]   **Gang Chen**[1]   **Junbo Zhao**[1*]
[1]Zhejiang University   [2]University of Wisconsin-Madison
[3]Chongqing University   [4]RIKEN

## Abstract

Partial label learning (PLL) is an important problem that allows each training example to be labeled with a coarse candidate set, which well suits many real-world data annotation scenarios with label ambiguity. Despite the promise, the performance of PLL often lags behind the supervised counterpart. In this work, we bridge the gap by addressing two key research challenges in PLL—representation learning and label disambiguation—in one coherent framework. Specifically, our proposed framework PiCO consists of a contrastive learning module along with a novel class prototype-based label disambiguation algorithm. PiCO produces closely aligned representations for examples from the same classes and facilitates label disambiguation. Theoretically, we show that these two components are mutually beneficial, and can be rigorously justified from an expectation-maximization (EM) algorithm perspective. Extensive experiments demonstrate that PiCO significantly outperforms the current state-of-the-art approaches in PLL and even achieves comparable results to fully supervised learning. Code and data available: https://github.com/hbzju/PiCO.

## 1 Introduction

The training of modern deep neural networks typically requires massive labeled data, which imposes formidable obstacles in data collection. Of a particular challenge, data annotation in the real-world can naturally be subject to inherent label ambiguity and noise. For example, as shown in Figure 1, identifying an Alaskan Malamute from a Siberian Husky can be difficult for a human annotator. The issue of labeling ambiguity is prevalent yet often overlooked in many applications, such as web mining (Luo & Orabona, 2010) and automatic image annotation (Chen et al., 2018). This gives rise to the importance of *partial label learning* (PLL) (Hüllermeier & Beringer, 2006; Cour et al., 2011), where each training example is equipped with a set of candidate labels instead of the exact ground-truth label. This stands in contrast to its supervised counterpart where one label must be chosen as the "gold". Arguably, the PLL problem is deemed more common and practical in various situations due to its relatively lower cost to annotations.

Despite the promise, a core challenge in PLL is label disambiguation, i.e., identifying the ground-truth label from the candidate label set. Existing methods typically require a good feature representation (Liu & Dietterich, 2012; Zhang et al., 2016; Lyu et al., 2021), and operate under the assumption that data points closer in the feature space are more likely to share the same ground-truth label. However, the reliance on representations has led to a non-trivial dilemma—the inherent label uncertainty can undesirably manifest in the representation learning process—the quality of which may, in turn, prevent effective label disambiguation. To date, few efforts have been made to resolve this.

This paper bridges the gap by reconciling the intrinsic tension between the two highly dependent problems—representation learning and label disambiguation—in one coherent and synergistic framework. Our framework, **Parti**al label learning with **CO**ntrastive label disambiguation (dubbed **PiCO**), produces closely aligned representations for examples from the same classes and facilitates label disambiguation. Specifically, PiCO encapsulates two key components. First, we leverage contrastive learning (CL) (Khosla et al., 2020) to partial label learning, which is unexplored in previous

---

*Correspondence to j.zhao@zju.edu.cn.

PLL literature. To mitigate the key challenge of constructing positive pairs, we employ the classifier's output and generate pseudo positive pairs for contrastive comparison (Section 3.1). Second, based on the learned embeddings, we propose a novel prototype-based label disambiguation strategy (Section 3.2). Key to our method, we gradually update the pseudo target for classification, based on the closest class prototype. By alternating the two steps above, PiCO converges to a solution with a highly distinguishable representation for accurate classification. Empirically, PiCO establishes *state-of-the-art* performance on three benchmark datasets, outperforming the baselines by a significant margin (Section 4) and obtains results that are competitive with *fully supervised learning*.

Theoretically, we demonstrate that our contrastive representation learning and prototype-based label disambiguation are mutually beneficial, and can be rigorously interpreted from an Expectation-Maximization (EM) algorithm perspective (Section 5). First, the refined pseudo labeling improves contrastive learning by selecting pseudo positive examples accurately. This can be analogous to the E-step, where we utilize the classifier's output to assign each data example to one label-specific cluster. Second, better contrastive performance in turn improves the quality of representations and thus the effectiveness of label disambiguation. This can be reasoned from an M-step perspective, where the contrastive loss partially maximizes the likelihood by clustering similar data examples. Finally, the training data will be mapped to a mixture of von Mises-Fisher distributions on the unit hypersphere, which facilitates label disambiguation by using the component-specific label.

Our **main contributions** are summarized as follows:

1 (Methodology). To the best of our knowledge, our paper pioneers the exploration of contrastive learning for partial label learning and proposes a novel framework termed PiCO. As an integral part of our algorithm, we also introduce a new prototype-based label disambiguation mechanism, that leverages the contrastively learned embeddings.

2 (Experiments). Empirically, our proposed PiCO framework establishes the *state-of-the-art* performance on three PLL tasks. Moreover, we make the first attempt to conduct experiments on fine-grained classification datasets, where we show classification performance improvement by up to **9.61**% compared with the best baseline on the CUB-200 dataset.

3 (Theory). We theoretically interpret our framework from the expectation-maximization perspective. Our derivation is also generalizable to other CL methods and shows the *alignment* property in CL (Wang & Isola, 2020) mathematically equals the M-step in center-based clustering algorithms.

## 2 BACKGROUND

The problem of *partial label learning* (PLL) is defined using the following setup. Let $\mathcal{X}$ be the input space, and $\mathcal{Y} = \{1, 2, ..., C\}$ be the output label space. We consider a training dataset $\mathcal{D} = \{(\boldsymbol{x}_i, Y_i)\}_{i=1}^{n}$, where each tuple comprises of an image $\boldsymbol{x}_i \in \mathcal{X}$ and a *candidate label set* $Y_i \subset \mathcal{Y}$. Identical to the supervised learning setup, the goal of PLL is to obtain a functional mapping that predicts the one true label associated with the input. Yet differently, the PLL setup bears significantly more uncertainty in the label space. A basic assumption of PLL is that the ground-truth label $y_i$ is concealed in its candidate set, i.e., $y_i \in Y_i$, and is invisible to the learner. For this reason, the learning process can suffer from inherent ambiguity, compared with the supervised learning task with explicit ground-truth.

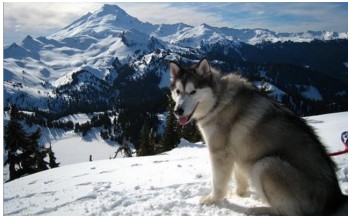

A dog image $\boldsymbol{x}_i$ with
$Y_i = \{\text{Husky}, \underline{\text{Malamute}}, \text{Samoyed}\}$

Figure 1: An input image with three candidate labels, where the ground-truth is `Malamute`.

The key challenge of PLL is to identify the ground-truth label from the candidate label set. During training, we assign each image $\boldsymbol{x}_i$ a normalized vector $\boldsymbol{s}_i \in [0, 1]^C$ as the *pseudo target*, whose entries denote the probability of labels being the ground-truth. The total probability mass of 1 is allocated among candidate labels in $Y_i$. Note that $\boldsymbol{s}_i$ will be updated during the training procedure. Ideally, $\boldsymbol{s}_i$ should put more probability mass on the (unknown) ground-truth label $y_i$ over the course of training. We train a classifier $f : \mathcal{X} \rightarrow [0, 1]^C$ using cross-entropy loss, with $\boldsymbol{s}_i$ being the target prediction. The per-sample loss is given by:

$$\mathcal{L}_{\text{cls}}(f; \boldsymbol{x}_i, Y_i) = \sum_{j=1}^{C} -s_{i,j} \log(f^j(\boldsymbol{x}_i)) \quad \text{s.t.} \quad \sum_{j \in Y_i} s_{i,j} = 1 \text{ and } s_{i,j} = 0, \forall j \notin Y_i, \quad (1)$$

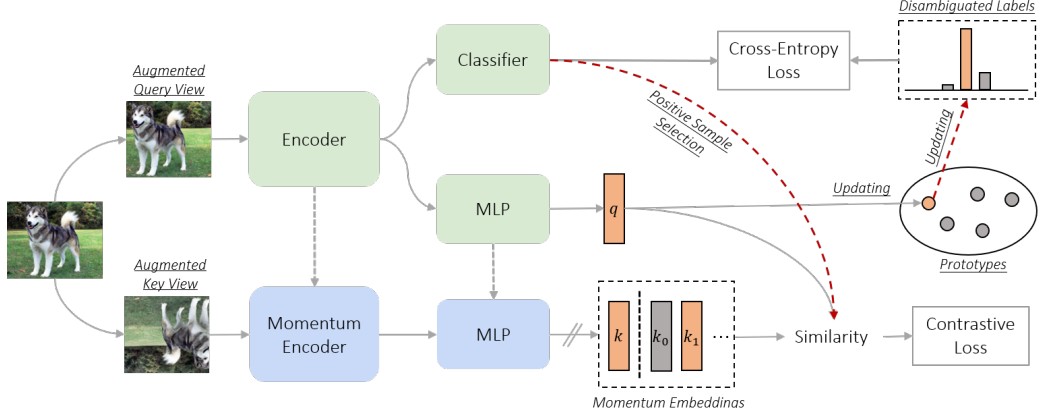

Figure 2: Illustration of PiCO. The classifier's output is used to determine the positive peers for contrastive learning. The contrastive prototypes are then used to gradually update the pseudo target. The momentum embeddings are maintained by a queue structure. '//' means stop gradient.

where $j$ denotes the indices of labels. $s_{i,j}$ denotes the $j$-th pseudo target of $\boldsymbol{x}_i$. Here $f$ is the softmax output of the networks and we denote $f^j$ as its $j$-th entry. In the remainder of this paper, we omit the sample index $i$ when the context is clear. We proceed by describing our proposed framework.

## 3 METHOD

In this section, we describe our novel **P**art**i**al label learning with **CO**ntrastive label disambiguation (PiCO) framework in detail. In a nutshell, PiCO comprises two key components tackling the representation quality (Section 3.1) and label ambiguity respectively (Section 3.2). The two components systematically work as a whole and reciprocate each other. We further rigorously provide a theoretical interpretation of PiCO from an EM perspective in Section 5.

### 3.1 CONTRASTIVE REPRESENTATION LEARNING FOR PLL

The uncertainty in the label space posits a unique obstacle for learning effective representations. In PiCO, we couple the classification loss in Eq. (1) with a contrastive term that facilitates a clustering effect in the embedding space. While contrastive learning has been extensively studied in recent literature, it remains untapped in the domain of PLL. The main challenge lies in the construction of a positive sample set. In conventional supervised CL frameworks, the positive sample pairs can be easily drawn according to the ground-truth labels (Khosla et al., 2020). However, this is not straightforward in the setting of PLL.

**Training Objective.** To begin with, we describe the standard contrastive loss term. We adopt the most popular setups by closely following MoCo (He et al., 2020) and SupCon (Khosla et al., 2020). Given each sample $(\boldsymbol{x}, Y)$, we generate two views—a query view and a key view—by way of randomized data augmentation $\mathrm{Aug}(\boldsymbol{x})$. The two images are then fed into a query network $g(\cdot)$ and a key network $g'(\cdot)$, yielding a pair of $L_2$-normalized embeddings $\boldsymbol{q} = g(\mathrm{Aug}_q(\boldsymbol{x}))$ and $\boldsymbol{k} = g'(\mathrm{Aug}_k(\boldsymbol{x}))$. In implementations, the query network shares the same convolutional blocks as the classifier, followed by a prediction head (see Figure 2). Following MoCo, the key network uses a momentum update with the query network. We additionally maintain a queue storing the most current key embeddings $\boldsymbol{k}$, and we update the queue chronologically. To this end, we have the following contrastive embedding pool:

$$A = B_q \cup B_k \cup \text{queue}, \tag{2}$$

where $B_q$ and $B_k$ are vectorial embeddings corresponding to the query and key views of the current mini-batch. Given an example $\boldsymbol{x}$, the per-sample contrastive loss is defined by contrasting its query embedding with the remainder of the pool $A$,

$$\mathcal{L}_{\text{cont}}(g; \boldsymbol{x}, \tau, A) = -\frac{1}{|P(\boldsymbol{x})|} \sum_{\boldsymbol{k}_+ \in P(\boldsymbol{x})} \log \frac{\exp(\boldsymbol{q}^\top \boldsymbol{k}_+/\tau)}{\sum_{\boldsymbol{k}' \in A(\boldsymbol{x})} \exp(\boldsymbol{q}^\top \boldsymbol{k}'/\tau)}, \tag{3}$$

where $P(\boldsymbol{x})$ is the positive set and $A(\boldsymbol{x}) = A\backslash\{\boldsymbol{q}\}$. $\tau \geq 0$ is the temperature.

**Positive Set Selection.** As mentioned earlier, the crucial challenge is how to construct the positive set $P(\boldsymbol{x})$. We propose utilizing the predicted label $\tilde{y} = \arg\max_{j\in Y} f^j(\text{Aug}_q(\boldsymbol{x}))$ from the classifier. Note that we restrict the predicted label to be in the candidate set $Y$. The positive examples are then selected as follows,

$$P(\boldsymbol{x}) = \{\boldsymbol{k}'|\boldsymbol{k}' \in A(\boldsymbol{x}), \tilde{y}' = \tilde{y}\}. \tag{4}$$

where $\tilde{y}'$ is the predicted label for the corresponding training example of $\boldsymbol{k}'$. For computational efficiency, we also maintain a label queue to store past predictions. In other words, we define the positive set of $\boldsymbol{x}$ to be those examples carrying the same *approximated* label prediction $\tilde{y}$. Despite its simplicity, we show that our selection strategy can be theoretically justified (Section 5) and also lead to superior empirical results (Section 4). Note that more sophisticated selection strategies can be explored, for which we discuss in Appendix B.4. Putting it all together, we jointly train the classifier as well as the contrastive network. The overall loss function is:

$$\mathcal{L} = \mathcal{L}_{\text{cls}} + \lambda\mathcal{L}_{\text{cont}}. \tag{5}$$

Still, our goal of learning high-quality representation by CL relies on accurate classifier prediction for positive set selection, which remains unsolved in the presence of label ambiguity. To this end, we further propose a novel label disambiguation mechanism based on contrastive embeddings and show that these two components are mutually beneficial.

## 3.2 PROTOTYPE-BASED LABEL DISAMBIGUATION

As we mentioned (and later theoretically prove in Section 5), the contrastive loss poses a clustering effect in the embedding space. As a collaborative algorithm, we introduce our novel prototype-based label disambiguation strategy. Importantly, we keep a *prototype* embedding vector $\boldsymbol{\mu}_c$ corresponding to each class $c \in \{1, 2, ..., C\}$, which can be deemed as a set of representative embedding vectors. Categorically, a naive version of the pseudo target assignment is to find the nearest prototype of the current embedding vector. Notably this primitive resembles a clustering step. We additionally soften this hard label assignment version by using a moving-average style formula. To this end, we may posit intuitively that the employment of the prototype builds a connection with the clustering effect in the embedding space brought by the contrastive term (Section 3.1). We provide a more rigorous justification in Section 5.

**Pseudo Target Updating.** We propose a softened and moving-average style strategy to update the pseudo targets. Specifically, we first initialize the pseudo targets with a uniform distribution, $s_j = \frac{1}{|Y|}\mathbb{I}(j \in Y)$. We then iteratively update it by the following moving-average mechanism,

$$\boldsymbol{s} = \phi\boldsymbol{s} + (1-\phi)\boldsymbol{z}, \quad z_c = \begin{cases} 1 & \text{if } c = \arg\max_{j\in Y} \boldsymbol{q}^\top\boldsymbol{\mu}_j, \\ 0 & \text{else} \end{cases} \tag{6}$$

where $\phi \in (0, 1)$ is a positive constant, and $\boldsymbol{\mu}_j$ is a *prototype* corresponding to the $j$-th class. The intuition is that fitting uniform pseudo targets results in a good initialization for the classifier since the contrastive embeddings are less distinguishable at the beginning. The moving-average style strategy then smoothly updates the pseudo targets towards the correct ones, and meanwhile ensures stable dynamics of training; see Appendix B.2. With more rigorous validation provided later in Section 5, we provide an explanation for the prototype as follows: (i)-for a given input $\boldsymbol{x}$, the closest prototype is indicative of its ground-truth class label. At each step, $\boldsymbol{s}$ has the tendency to slightly move toward the one-hot distribution defined by $\boldsymbol{z}$ based on Eq. (6); (ii)-if an example consistently points to one prototype, the pseudo target $\boldsymbol{s}$ can converge (almost) to a one-hot vector with the least ambiguity.

**Prototype Updating.** The most canonical way to update the prototype embeddings is to compute it in every iteration of training. However, this would extract a heavy computational toll and in turn cause unbearable training latency. As a result, we update the class-conditional prototype vector similarly in a moving-average style:

$$\boldsymbol{\mu}_c = \text{Normalize}(\gamma\boldsymbol{\mu}_c + (1-\gamma)\boldsymbol{q}), \quad \text{if } c = \arg\max_{j\in Y} f^j(\text{Aug}_q(\boldsymbol{x}))), \tag{7}$$

where the momentum prototype $\boldsymbol{\mu}_c$ of class $c$ is defined by the moving-average of the normalized query embeddings $\boldsymbol{q}$ whose predicted class conforms to $c$. $\gamma$ is a tunable hyperparameter.

Table 1: Accuracy comparisons on benchmark datasets. Bold indicates superior results. Notably, PiCO achieves comparable results to the fully supervised learning (less than 1% in accuracy with $\approx 1$ false candidate).

| Dataset | Method | $q = 0.1$ | $q = 0.3$ | $q = 0.5$ |
|---|---|---|---|---|
| | PiCO (ours) | **94.39** $\pm 0.18\%$ | **94.18** $\pm 0.12\%$ | **93.58** $\pm 0.06\%$ |
| | LWS | $90.30 \pm 0.60\%$ | $88.99 \pm 1.43\%$ | $86.16 \pm 0.85\%$ |
| | PRODEN | $90.24 \pm 0.32\%$ | $89.38 \pm 0.31\%$ | $87.78 \pm 0.07\%$ |
| CIFAR-10 | CC | $82.30 \pm 0.21\%$ | $79.08 \pm 0.07\%$ | $74.05 \pm 0.35\%$ |
| | MSE | $79.97 \pm 0.45\%$ | $75.64 \pm 0.28\%$ | $67.09 \pm 0.66\%$ |
| | EXP | $79.23 \pm 0.10\%$ | $75.79 \pm 0.21\%$ | $70.34 \pm 1.32\%$ |
| | Fully Supervised | | $94.91 \pm 0.07\%$ | |

| Dataset | Method | $q = 0.01$ | $q = 0.05$ | $q = 0.1$ |
|---|---|---|---|---|
| | PiCO (ours) | **73.09** $\pm 0.34\%$ | **72.74** $\pm 0.30\%$ | **69.91** $\pm 0.24\%$ |
| | LWS | $65.78 \pm 0.02\%$ | $59.56 \pm 0.33\%$ | $53.53 \pm 0.08\%$ |
| | PRODEN | $62.60 \pm 0.02\%$ | $60.73 \pm 0.03\%$ | $56.80 \pm 0.29\%$ |
| CIFAR-100 | CC | $49.76 \pm 0.45\%$ | $47.62 \pm 0.08\%$ | $35.72 \pm 0.47\%$ |
| | MSE | $49.17 \pm 0.05\%$ | $46.02 \pm 1.82\%$ | $43.81 \pm 0.49\%$ |
| | EXP | $44.45 \pm 1.50\%$ | $41.05 \pm 1.40\%$ | $29.27 \pm 2.81\%$ |
| | Fully Supervised | | $73.56 \pm 0.10\%$ | |

## 3.3 SYNERGY BETWEEN CONTRASTIVE LEARNING AND LABEL DISAMBIGUATION

While seemingly separated from each other, the two key components of PiCO work in a collaborative fashion. First, as the contrastive term favorably manifests a clustering effect in the embedding space, the label disambiguation module further leverages via setting more precise prototypes. Second, a set of well-polished label disambiguation results may, in turn, reciprocate the positive set construction which serves as a crucial part in the contrastive learning stage. The entire training process converges when the two components perform satisfactorily. We further rigorously draw a resemblance of PiCO with a classical EM-style clustering algorithm in Section 5. Our experiments, particularly the ablation study displayed in Section 4.3, further justify the mutual dependency of the synergy between the two components. The pseudo-code of our complete algorithm is shown in Appendix C.

## 4 EXPERIMENTS

### 4.1 SETUP

**Datasets and Baselines.** First, we evaluate PiCO on two commonly used benchmarks — CIFAR-10 and CIFAR-100 (Krizhevsky et al., 2009). Adopting the identical experimental settings in previous work (Lv et al., 2020; Wen et al., 2021), we generate partially labeled datasets by flipping negative labels $\bar{y} \neq y$ to false positive labels with a probability $q = P(\bar{y} \in Y | \bar{y} \neq y)$. In other words, all $C - 1$ negative labels have a uniform probability to be false positive and we aggregate the flipped ones with the ground-truth to form the candidate set. We consider $q \in \{0.1, 0.3, 0.5\}$ for CIFAR-10 and $q \in \{0.01, 0.05, 0.1\}$ for CIFAR-100. In Section 4.4, we further evaluate our method on fine-grained classification tasks, where label disambiguation can be more challenging.

We choose the five best-performed partial label learning algorithms to date: 1) LWS (Wen et al., 2021) weights the risk function by means of a trade-off between losses on candidate labels and the remaining; 2) PRODEN (Lv et al., 2020) iteratively updates the latent label distribution in a self-training style; 3) CC (Feng et al., 2020b) is a classifier-consistent method that assumes set-level uniform data generation process; 4) MSE and EXP (Feng et al., 2020a) are two simple baselines that adopt mean square error and exponential loss as the risks. The hyperparameters are tuned according to the original methods. The detailed implementation of our method PiCO is presented in Appendix B.1. For all experiments, we report the mean and standard deviation based on 5 independent runs (with different random seeds).

### 4.2 MAIN EMPIRICAL RESULTS

**PiCO achieves SOTA results.** As shown in Table 1, PiCO significantly outperforms all the rivals by a significant margin on all datasets. Specifically, on CIFAR-10 dataset, we improve upon the best baseline by **4.09**%, **4.80**%, and **5.80**% where $q$ is set to $0.1, 0.3, 0.5$ respectively. Moreover, PiCO consistently achieves superior results as the size of the candidate set increases, while the baselines demonstrate a significant performance drop. Besides, it is worth pointing out that previous

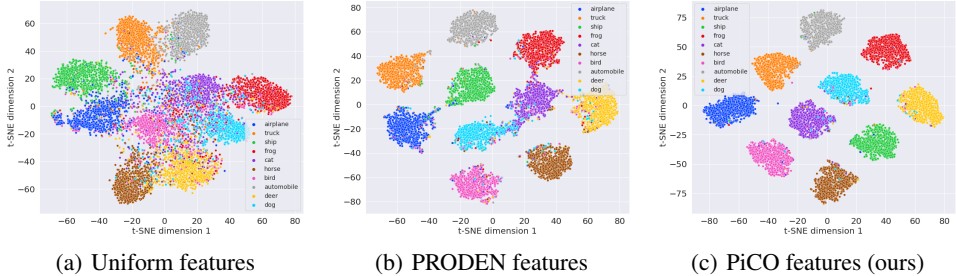

(a) Uniform features  (b) PRODEN features  (c) PiCO features (ours)

Figure 3: T-SNE visualization of the image representation on CIFAR-10 with $q = 0.5$. Different colors represent the corresponding classes.

works (Lv et al., 2020; Wen et al., 2021) are typically evaluated on datasets with a small label space ($C = 10$). We challenge this by showing additional results on CIFAR-100. When $q = 0.1$, all the baselines fail to obtain satisfactory performance, whereas PiCO remains competitive. Finally, we observe that PiCO achieves results that are comparable to the *fully supervised contrastive learning model*, showing that disambiguation is sufficiently accomplished. The comparison highlights the superiority of our label disambiguation strategy.

**PiCO learns more distinguishable representations.** We visualize the image representation produced by the feature encoder using t-SNE (Van der Maaten & Hinton, 2008) in Figure 3. Different colors represent different ground-truth class labels. We use the CIFAR-10 dataset with $q = 0.5$. We contrast the t-SNE embeddings of three approaches: (a) a model trained with uniform pseudo targets, i.e., $s_j = 1/|Y|$ ($j \in Y$), (b) the best baseline PRODEN, and (c) our method PiCO. We can observe that the representation of the uniform model is indistinguishable since its supervision signals suffer from high uncertainty. The features of PRODEN are improved, yet with some class overlapping (e.g., blue and purple). In contrast, PiCO produces well-separated clusters and more distinguishable representations, which validates its effectiveness in learning high-quality representation.

## 4.3 ABLATION STUDIES

In this section, we present part of our ablation results to show the effectiveness of PiCO. We refer readers to Appendix B.2 for more ablation experiments.

**Effect of $\mathcal{L}_{\text{cont}}$ and label disambiguation.** We ablate the contributions of two key components of PiCO: contrastive learning and prototype-based label disambiguation. In particular, we compare PiCO with two variants: 1) *PiCO w/o disambiguation* which keeps the pseudo target as uniform $1/|Y|$; and 2) *PiCO w/o $\mathcal{L}_{cont}$*, which further removes the contrastive learning and only trains a classifier with uniform pseudo targets. From Table 2, we can observe that variant 1 substantially outperforms variant 2 (e.g., +8.04% on CIFAR-10), which signifies the importance of contrastive learning for producing better representations. Moreover, with label disambiguation, PiCO obtains results close to *fully supervised setting*, which verifies the ability of PiCO in identifying the ground-truth.

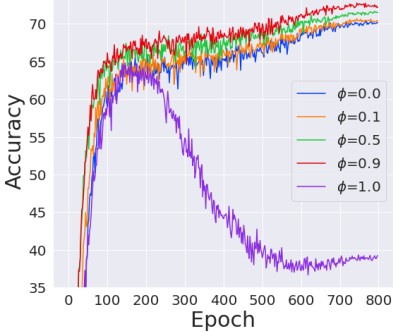

Figure 4: Performance of PiCO with varying $\phi$ on CIFAR-100 ($q = 0.05$).

**Different disambiguation strategy.** Based on the contrastive prototypes, various strategies can be used to disambiguate the labels, which corresponds to the E-step in our theoretical analysis. We choose the following variants: 1) *One-hot Prototype* always assigns a one-hot pseudo target $s = z$ by using the nearest prototype ($\phi = 0$); 2) *Soft Prototype Probs* follows (Li et al., 2021a) and uses a soft class probability $s_i = \frac{\exp(q^\top \mu_i/\tau)}{\sum_{j \in Y} \exp(q^\top \mu_j/\tau)}$ as the pseudo target ($\phi = 0$); 3) *MA Soft Prototype Probs* gradually updates pseudo target from uniform by using the soft probabilities in a moving-average style. From Table 2, we can see that directly using either soft or hard prototype-based label assignment leads to competitive results. This corroborates our theoretical analysis in Section 5, since center-based class probability estimation is common in clustering algorithms. However, *MA Soft Prototype Probs* displays degenerated performance, suggesting soft label assignment is less reliable in identifying the ground-truth. Finally, PiCO outperforms the best variant by $\approx 2\%$ in accuracy on both datasets, showing the superiority of our label disambiguation strategy.

Table 2: Ablation study on CIFAR-10 with $q = 0.5$ and CIFAR-100 with $q = 0.05$.

| Ablation | $\mathcal{L}_{\text{cont}}$ | Label Disambiguation | CIFAR-10 ($q = 0.5$) | CIFAR-100 ($q = 0.05$) |
|---|---|---|---|---|
| PiCO | ✓ | Ours | **93.58** | **72.74** |
| PiCO w/o Disambiguation | ✓ | Uniform Pseudo Target | 84.50 | 64.11 |
| PiCO w/o $\mathcal{L}_{\text{cont}}$ | ✗ | Uniform Pseudo Target | 76.46 | 56.87 |
| PiCO with $\phi = 0$ | ✓ | Soft Prototype Probs | 91.60 | 71.07 |
| PiCO with $\phi = 0$ | ✓ | One-hot Prototype | 91.41 | 70.10 |
| PiCO | ✓ | MA Soft Prototype Probs | 81.67 | 63.75 |

**Effect of moving-average factor** $\phi$. We then explore the effect of pseudo target updating factor $\phi$ on PiCO performance. Figure 4 shows the learning curves of PiCO on CIFAR-100 ($q = 0.05$). We can see that the best result is achieved at $\phi = 0.9$ and the performance drops when $\phi$ takes a smaller value, particularly on the early stage. When $\phi = 0$, PiCO obtains a competitive result but is much lower than $\phi = 0.9$. This confirms that trusting the uniform pseudo targets at the early stage is crucial in obtaining superior performance. At the other extreme value $\phi = 1$, uniform pseudo targets are used, and PiCO demonstrates a degenerated performance and severe overfitting phenomena. In general, PiCO performs well when $\phi \approx 0.9$.

## 4.4 FURTHER EXTENSION: FINE-GRAINED PARTIAL LABEL LEARNING

Recall the dog example highlighted in Section 2, where semantically similar classes are more likely to cause label ambiguity. It begs the question of whether PiCO is effective on the challenging fine-grained image classification tasks. To verify this, we conduct experiments on two datasets: 1) CUB-200 dataset (Welinder et al., 2010) contains 200 bird species; 2) CIFAR-100 with hierarchical labels (CIFAR-100-H), where we generate candidate labels that belong to the same superclass[1]. We set $q = 0.05$ for CUB-200 and $q = 0.5$ for CIFAR-100 with hi-

Table 3: Accuracy comparisons on fine-grained classification datasets.

| Method | CUB-200 ($q = 0.05$) | CIFAR-100-H ($q = 0.5$) |
|---|---|---|
| PiCO | **72.17** $\pm$ 0.72% | **72.04** $\pm$ 0.31% |
| LWS | 39.74 $\pm$ 0.47% | 57.25 $\pm$ 0.02% |
| PRODEN | 62.56 $\pm$ 0.10% | 60.89 $\pm$ 0.03% |
| CC | 55.61 $\pm$ 0.02% | 42.60 $\pm$ 0.11% |
| MSE | 22.07 $\pm$ 2.36% | 39.52 $\pm$ 0.28% |
| EXP | 9.44 $\pm$ 2.32% | 35.08 $\pm$ 1.71% |

erarchical labels. In Table 3, we compare PiCO with baselines, where PiCO outperforms the best method PRODEN by a large margin (**+9.61**% on CUB-200 and **+11.15**% on CIFAR-100-H). Our results validate the effectiveness of our framework, even in the presence of strong label ambiguity.

## 5 WHY PiCO IMPROVES PARTIAL LABEL LEARNING?

In this section, we provide theoretical justification on why the contrastive prototypes help disambiguate the ground-truth label. We show that the *alignment* property in contrastive learning (Wang & Isola, 2020) intrinsically minimizes the intraclass covariance in the embedding space, which coincides with the objective of classical clustering algorithms. It motivates us to interpret PiCO through the lens of the expectation-maximization algorithm. To see this, we consider an ideal setting: in each training step, all data examples are accessible and the augmentation copies are also included in the training set, i.e., $A = \mathcal{D}$. Then, the contrastive loss is calculated as,

$$
\begin{aligned}
\tilde{\mathcal{L}}_{\text{cont}}(g; \tau, \mathcal{D}) &= \frac{1}{n} \sum_{\boldsymbol{x} \in \mathcal{D}} \left\{ -\frac{1}{|P(\boldsymbol{x})|} \sum_{\boldsymbol{k}_+ \in P(\boldsymbol{x})} \log \frac{\exp(\boldsymbol{q}^\top \boldsymbol{k}_+/\tau)}{\sum_{\boldsymbol{k}' \in A(\boldsymbol{x})} \exp(\boldsymbol{q}^\top \boldsymbol{k}'/\tau)} \right\} \\
&= \underbrace{\frac{1}{n} \sum_{\boldsymbol{x} \in \mathcal{D}} \left\{ -\frac{1}{|P(\boldsymbol{x})|} \sum_{\boldsymbol{k}_+ \in P(\boldsymbol{x})} (\boldsymbol{q}^\top \boldsymbol{k}_+/\tau) \right\}}_{(a)} + \underbrace{\frac{1}{n} \sum_{\boldsymbol{x} \in \mathcal{D}} \left\{ \log \sum_{\boldsymbol{k}' \in A(\boldsymbol{x})} \exp(\boldsymbol{q}^\top \boldsymbol{k}'/\tau) \right\}}_{(b)}.
\end{aligned}
\tag{8}
$$

We focus on analyzing the first term (a), which is often dubbed as the *alignment* term (Wang & Isola, 2020). The main functionality of this term is to optimize the tightness of the clusters in the embedding space. In this work, we connect it with classical clustering algorithms. We first split the dataset to $C$ subsets $S_j \in \mathcal{D}_C$ ($1 \leq j \leq C$), where each subset contains examples possessing

---

[1] CIFAR-100 dataset consists of 20 superclasses, with 5 classes in each superclass.

the same predicted labels. In effect, our selection strategy in Eq. (4) constructs the positive set by selecting examples from the same subset. Therefore, we have,

$$
\begin{aligned}
\text{(a)} &= \frac{1}{n} \sum_{\boldsymbol{x} \in \mathcal{D}} \frac{1}{|P(\boldsymbol{x})|} \sum_{\boldsymbol{k}_+ \in P(\boldsymbol{x})} (\|\boldsymbol{q} - \boldsymbol{k}_+\|^2 - 2)/(2\tau) \\
&\approx \frac{1}{2\tau n} \sum_{S_j \in \mathcal{D}_C} \frac{1}{|S_j|} \sum_{\boldsymbol{x}, \boldsymbol{x}' \in S_j} \|g(\boldsymbol{x}) - g(\boldsymbol{x}')\|^2 + K \\
&= \frac{1}{\tau n} \sum_{S_j \in \mathcal{D}_C} \sum_{\boldsymbol{x} \in S_j} \|g(\boldsymbol{x}) - \boldsymbol{\mu}_j\|^2 + K,
\end{aligned}
\tag{9}
$$

where $K$ is a constant and $\boldsymbol{\mu}_j$ is the mean center of $S_j$. Here we approximate $\frac{1}{|S_j|} \approx \frac{1}{|S_j|-1} = \frac{1}{|P(\boldsymbol{x})|}$ since $n$ is usually large. We omitted the augmentation operation for simplicity. The *uniformity* term (b) can benefit information-preserving, and has been analyzed in (Wang & Isola, 2020).

We are now ready to interpret the PiCO algorithm as an expectation-maximization algorithm that maximizes the likelihood of a generative model. At the E-step, the classifier assigns each data example to one specific cluster. At the M-step, the contrastive loss concentrates the embeddings to their cluster mean direction, which is achieved by minimizing Eq. (9). Finally, the training data will be mapped to a mixture of von Mises-Fisher distributions on the unit hypersphere.

**EM Perspective.** Recall that the candidate label set is a noisy version of the ground-truth. To estimate the likelihood $P(Y_i, \boldsymbol{x}_i)$, we need to establish the relationship between the candidate and the ground-truth label. Following (Liu & Dietterich, 2012), we make a mild assumption,

**Assumption 1.** *All labels $y_i$ in the candidate label set have the same probability of generating $Y_i$, but no label outside of $Y_i$ can generate $Y_i$, i.e. $P(Y_i|y_i) = \hbar(Y_i)$ if $y_i \in Y_i$ else $0$. Here $\hbar(\cdot)$ is some function making it a valid probability distribution.*

Then, we show that the PiCO implicitly maximizes the likelihood as follows,

**E-Step.** First, we introduce some distributions over all examples and the candidates $\pi_i^j \geq 0$ ($1 \leq i \leq n, 1 \leq j \leq C$) such that $\pi_i^j = 0$ if $j \notin Y_i$ and $\sum_{j \in Y_i} \pi_i^j = 1$. Let $\theta$ be the parameters of $g$. Our goal is to maximize the likelihood below,

$$
\begin{aligned}
\operatorname*{argmax}_\theta \sum_{i=1}^n \log P(Y_i, \boldsymbol{x}_i|\theta) &= \operatorname*{argmax}_\theta \sum_{i=1}^n \log \sum_{y_i \in Y_i} P(\boldsymbol{x}_i, y_i|\theta) + \sum_{i=1}^n \log(\hbar(Y_i)) \\
&= \operatorname*{argmax}_\theta \sum_{i=1}^n \log \sum_{y_i \in Y_i} \pi_i^{y_i} \frac{P(\boldsymbol{x}_i, y_i|\theta)}{\pi_i^{y_i}} \\
&\geq \operatorname*{argmax}_\theta \sum_{i=1}^n \sum_{y_i \in Y_i} \pi_i^{y_i} \log \frac{P(\boldsymbol{x}_i, y_i|\theta)}{\pi_i^{y_i}}.
\end{aligned}
\tag{10}
$$

The last step of the derivation uses Jensen's inequality. By using the fact that $\log(\cdot)$ function is concave, the inequality holds with equality when $\frac{P(\boldsymbol{x}_i, y_i|\theta)}{\pi_i^{y_i}}$ is some constant. Therefore,

$$
\pi_i^{y_i} = \frac{P(\boldsymbol{x}_i, y_i|\theta)}{\sum_{y_i \in Y_i} P(\boldsymbol{x}_i, y_i|\theta)} = \frac{P(\boldsymbol{x}_i, y_i|\theta)}{\sum_{y_i=1}^C P(\boldsymbol{x}_i, y_i|\theta)} = \frac{P(\boldsymbol{x}_i, y_i|\theta)}{P(\boldsymbol{x}_i|\theta)} = P(y_i|\boldsymbol{x}_i, \theta),
\tag{11}
$$

which is the posterior class probability. In PiCO, it is estimated by using the classifier's output.

To estimate $P(y_i|\boldsymbol{x}_i, \theta)$, classical unsupervised clustering methods intuitively assign the data examples to the cluster centers, e.g. k-means. As in the supervised learning setting, we can directly use the ground-truth. However, under the setting of PLL, the supervision signals are situated between the supervised and unsupervised setups. Based on empirical findings, the candidate labels are more reliable for posterior estimation at the beginning; yet alongside the training process, the prototypes tend to become more trustful. This empirical observation has motivated us to update the pseudo targets in a moving-average style. Thereby, we have a good initialization in estimating class posterior, and it will be smoothly refined during the training procedure. This is verified in our empirical studies; see Section 4.3 and Appendix B.2. Finally, we take one-hot prediction $\tilde{y}_i = \arg\max_{j \in Y} f^j(\boldsymbol{x}_i)$ since each example inherently belongs to exactly one label and hence, we have $\pi_i^j = \mathbb{I}(\tilde{y}_i = j)$.

**M-Step.** At this step, we aim at maximizing the likelihood under the assumption that the posterior class probability is known. We show that under mild assumptions, minimizing Eq. (9) also maximizes a lower bound of likelihood in Eq. (10).

**Theorem 1.** *Assume data from the same class in the contrastive output space follow a d-variate von Mises-Fisher (vMF) distribution whose probabilistic density is given by $f(\boldsymbol{x}|\bar{\boldsymbol{\mu}}_i, \kappa) = c_d(\kappa)e^{\kappa\bar{\boldsymbol{\mu}}_i^\top g(\boldsymbol{x})}$, where $\bar{\boldsymbol{\mu}}_i = \boldsymbol{\mu}_i/||\boldsymbol{\mu}_i||$ is the mean direction, $\kappa$ is the concentration parameter, and $c_d(\kappa)$ is the normalization factor. We further assume a uniform class prior $P(y_i = j) = 1/C$. Let $n_j = |S_j|$. Then, optimizing Eq. (9) and Eq. (10) equal to maximize $R_1$ and $R_2$ below, respectively.*

$$R_1 = \sum\nolimits_{S_j \in \mathcal{D}_C} \frac{n_j}{n}||\boldsymbol{\mu}_j||^2 \leq \sum\nolimits_{S_j \in \mathcal{D}_C} \frac{n_j}{n}||\boldsymbol{\mu}_j|| = R_2. \tag{12}$$

The proof can be found in Appendix A. Theorem 1 indicates that minimizing Eq. (9) also maximizes a lower bound of the likelihood in Eq. (10). The lower bound is tight when $||\boldsymbol{\mu}_j||$ is close to 1, which in effect means a strong intraclass concentration on the hypersphere. Intuitively, when the hypothesis space is rich enough, it is possible to achieve a low intraclass covariance in the Euclidean space, resulting in a large norm of the mean vector $||\boldsymbol{\mu}_j||$. Then, normalized embeddings in the hypersphere also have an intraclass concentration in a strong sense, because a large $||\boldsymbol{\mu}_j||$ also results in a large $\kappa$ (Banerjee et al., 2005). Regarding the visualized representation in Figure 3, we note that PiCO is indeed able to learn compact clusters. Therefore, we have that minimizing the contrastive loss also partially maximizes the likelihood defined in Eq. (10).

## 6 RELATED WORKS

**Partial Label Learning** (PLL) allows each training example to be annotated with a candidate label set, in which the ground-truth is guaranteed to be included. The most intuitive solution is average-based methods (Hüllermeier & Beringer, 2006; Cour et al., 2011; Zhang & Yu, 2015), which treat all candidates equally. However, the key and obvious drawback is that the predictions can be severely misled by false positive labels. To disambiguate the ground-truth from the candidates, identification-based methods (Jin & Ghahramani, 2002), which regard the ground-truth as a latent variable, have recently attracted increasing attention; representative approaches include maximum margin-based methods (Nguyen & Caruana, 2008; Wang et al., 2020), graph-based methods (Zhang et al., 2016; Wang et al., 2019; Xu et al., 2019; Lyu et al., 2021), and clustering-based approaches (Liu & Dietterich, 2012). Recently, self-training methods (Feng et al., 2020b; Lv et al., 2020; Wen et al., 2021) have achieved state-of-the-art results on various benchmark datasets, which disambiguate the candidate label sets by means of the model outputs themselves. But, few efforts have been made to learn high-quality representations to reciprocate label disambiguation.

**Contrastive Learning** (CL) (van den Oord et al., 2018; He et al., 2020) is a framework that learns discriminative representations through the use of instance similarity/dissimilarity. A plethora of works has explored the effectiveness of CL in unsupervised representation learning (van den Oord et al., 2018; Chen et al., 2020; He et al., 2020). Recently, Khosla et al. (2020) propose supervised contrastive learning (SCL), an approach that aggregates data from the same class as the positive set and obtains improved performance on various supervised learning tasks. The success of SCL has motivated a series of works to apply CL to a number of weakly supervised learning tasks, including noisy label learning (Li et al., 2021a; Wu et al., 2021), semi-supervised learning (Li et al., 2020; Zhang et al., 2021), etc. Despite promising empirical results, however, these works, lack theoretical understandings. Wang & Isola (2020) theoretically show that the CL favors *alignment* and *uniformity*, and thoroughly analyzed the properties of uniformity. But, to date, the terminology *alignment* remains confusing; we show it inherently maps data points to a mixture of vMF distributions.

## 7 CONCLUSION

In this work, we propose a novel partial label learning framework PiCO. The key idea is to identify the ground-truth from the candidate set by using contrastively learned embedding prototypes. Empirically, we conducted extensive experiments and show that PiCO establishes state-of-the-art performance. Our results are competitive with the fully supervised setting, where the ground-truth label is given explicitly. Theoretical analysis shows that PiCO can be interpreted from an EM-algorithm perspective. Applications of multi-class classification with ambiguous labeling can benefit from our method, and we anticipate further research in PLL to extend this framework to tasks beyond image classification. We hope our work will draw more attention from the community toward a broader view of using contrastive prototypes for partial label learning.

ACKNOWLEDGMENTS

HW, RX, GC and JZ are supported by the Key R&D Program of Zhejiang Province (Grant No. 2020C01024). JZ would also like to thank the Fundamental Research Funds for the Central Universities. YL is supported by Wisconsin Alumni Research Foundation (WARF), Facebook Research Award, and funding from Google Research. FL is supported by the National Natural Science Foundation of China (Grant No. 62106028).

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

## A  THEORETICAL ANALYSIS

**Derivation of Eq. (9).**  We provide the derivation of the second equality in Eq. (9). It suffices to show that LHS $= \frac{1}{2n_j} \sum_{\boldsymbol{x},\boldsymbol{x}' \in S_j} ||g(\boldsymbol{x}) - g(\boldsymbol{x}')||^2 = \sum_{\boldsymbol{x} \in S_j} ||g(\boldsymbol{x}) - \boldsymbol{\mu}_j||^2 =$ RHS. We have,

$$
\begin{aligned}
\text{LHS} &= \frac{1}{2n_j} \sum_{\boldsymbol{x} \in S_j} \sum_{\boldsymbol{x}' \in S_j} (||g(\boldsymbol{x})||^2 - 2g(\boldsymbol{x})^\top g(\boldsymbol{x}') + ||g(\boldsymbol{x}')||^2) \\
&= \frac{1}{n_j} \sum_{\boldsymbol{x} \in S_j} (n_j - g(\boldsymbol{x})^\top (\sum_{\boldsymbol{x}' \in S_j} g(\boldsymbol{x}'))) \\
&= \frac{1}{n_j} \sum_{\boldsymbol{x} \in S_j} (n_j - g(\boldsymbol{x})^\top (n_j \boldsymbol{\mu}_j))) \\
&= n_j - (\sum_{\boldsymbol{x} \in S_j} g(\boldsymbol{x}))^\top \boldsymbol{\mu}_j = n_j(1 - ||\boldsymbol{\mu}_j||^2).
\end{aligned}
\tag{13}
$$

On the other hand,

$$
\begin{aligned}
\text{RHS} &= \sum_{\boldsymbol{x} \in S_j} (||g(\boldsymbol{x})||^2 - 2g(\boldsymbol{x})^\top \boldsymbol{\mu}_j + ||\boldsymbol{\mu}_j||^2) \\
&= (n_j - 2(\sum_{\boldsymbol{x} \in S_j} g(\boldsymbol{x}))^\top \boldsymbol{\mu}_j + n_j ||\boldsymbol{\mu}_j||^2) \\
&= n_j(1 - ||\boldsymbol{\mu}_j||^2) = \text{LHS}.
\end{aligned}
\tag{14}
$$

**Proof of Theorem 1.**  By regarding $\pi_i^j$ as constants w.r.t $\theta$, we can get the following derivation from Eq. (10),

$$
\begin{aligned}
\arg\max_\theta \sum_{i=1}^n \sum_{y_i \in Y_i} \pi_i^{y_i} \log \frac{P(\boldsymbol{x}_i, y_i | \theta)}{\pi_i^{y_i}} &= \arg\max_\theta \sum_{i=1}^n \sum_{y_i \in Y_i} \pi_i^{y_i} \log P(\boldsymbol{x}_i | y_i, \theta) P(y_i) \\
&= \arg\max_\theta \sum_{i=1}^n \sum_{y_i \in Y_i} \mathbb{I}(\tilde{y}_i = y_i) \log P(\boldsymbol{x}_i | y_i, \theta) \\
&= \arg\max_\theta \sum_{S_j \in \mathcal{D}_C} \sum_{\boldsymbol{x} \in S_j} \log P(\boldsymbol{x} | y = j, \theta) \\
&= \arg\max_\theta \sum_{S_j \in \mathcal{D}_C} \sum_{\boldsymbol{x} \in S_j} (\kappa \bar{\boldsymbol{\mu}}_j^\top g(\boldsymbol{x}) + \log(c_d(\kappa))) \\
&= \arg\max_\theta \sum_{S_j \in \mathcal{D}_C} \frac{n_j}{n} ||\boldsymbol{\mu}_j||
\end{aligned}
\tag{15}
$$

where $n_j = |S_j|$. Here we ignore the constant factor $-\sum_{i=1}^n \sum_{y_i \in Y_i} \pi_i^{y_i} \log \pi_i^{y_i}$ w.r.t. $\theta$ in the first equality. In the last equality, we use the fact that $\boldsymbol{\mu}_j = \frac{1}{n_j} \sum_{\boldsymbol{x} \in S_j} g(\boldsymbol{x})$ and $\bar{\boldsymbol{\mu}}_j$ is the unit directional

vector of $\boldsymbol{\mu}_j$. From Eq. (9), we have that,

$$\arg\min_\theta \sum_{S_j \in \mathcal{D}_C} \sum_{\boldsymbol{x} \in S_j} ||g(\boldsymbol{x}) - \boldsymbol{\mu}_j||^2 = \arg\min_\theta \sum_{S_j \in \mathcal{D}_C} \sum_{\boldsymbol{x} \in S_j} (||g(\boldsymbol{x})||^2 - 2g(\boldsymbol{x})^\top \boldsymbol{\mu}_j + ||\boldsymbol{\mu}_j||^2)$$

$$= \arg\min_\theta \sum_{S_j \in \mathcal{D}_C} (n_j - n_j ||\boldsymbol{\mu}_j||^2)$$

$$= \arg\max_\theta \sum_{S_j \in \mathcal{D}_C} \frac{n_j}{n} ||\boldsymbol{\mu}_j||^2. \tag{16}$$

Note that the contrastive embeddings are distributed on the hypersphere $S^{d-1}$ and thus $||\boldsymbol{\mu}_j|| \in [0, 1]$. It can be directly derived that,

$$R_1 = \sum_{S_j \in \mathcal{D}_C} \frac{n_j}{n} ||\boldsymbol{\mu}_j||^2 \leq \sum_{S_j \in \mathcal{D}_C} \frac{n_j}{n} ||\boldsymbol{\mu}_j|| = R_2. \tag{17}$$

Therefore, maximizing the intraclass covariance in Eq. (9) is equivalent to maximizing a lower bound of the likelihood in Eq. (10). It can also be shown that $(R_2)^2 \leq R_1$ followed by the convexity of the squared function. Since $\arg\max R_2 = \arg\max(R_2)^2$, we have that the contrastive loss also maximizes an upper bound of $R_2$.

Unfortunately, there is no guarantee that the lower bound is tight without further assumptions. To see this, assume that we have two classes $y \in \{1, 2\}$ with equal-sized samples and their mean vectors have the norm of $||\boldsymbol{\mu}_1|| = 0.5$ and $||\boldsymbol{\mu}_1|| = 1$. We have that $R_1 = 0.625$ and $R_2 = 0.75$, which demonstrates a large discrepancy. It is interesting to see that when the norm of the mean vectors are the same, i.e. $||\boldsymbol{\mu}_j|| = ||\boldsymbol{\mu}_k||$ for all $1 \leq j \leq k \leq C$, we have $(R_2)^2 = R_1$ by the Jensen's inequality, making the upper bound tight. But, it is not a trivial condition.

To obtain a tight lower bound, what we need is a rich enough hypothesis space to achieve a low intraclass covariance in Eq. (9), and hence a large $R_1$. We show that it inherently produces compact vMF distributions. To see this, it should be noted that the concentration parameter $\kappa$ of a vMF distribution is given by the inverse of the ratio of Bessel functions of mean vector $\boldsymbol{\mu}_j$. Though it is not possible to obtain an analytic solution of $\kappa$, we have the following well-known approximation (Banerjee et al., 2005),

$$\kappa \approx \frac{d-1}{2(1 - ||\boldsymbol{\mu}_j||)}, \qquad \qquad \text{valid for large } ||\boldsymbol{\mu}_j||, \tag{18}$$

$$\kappa \approx d||\boldsymbol{\mu}_j|| \left( 1 + \frac{d}{d+2} ||\boldsymbol{\mu}_j||^2 + \frac{d^2(d+8)}{(d+2)^2(d+4)} ||\boldsymbol{\mu}_j||^4 \right), \qquad \text{valid for small } ||\boldsymbol{\mu}_j||. \tag{19}$$

The above approximations show that a larger norm of Euclidean's mean $\boldsymbol{\mu}_j$ typically leads to a stronger concentration on the hypersphere. By Eq. (16), we know that contrastive loss encourages a large norm of $\boldsymbol{\mu}_j$, and thus also tightly clusters the embeddings on the hypersphere; see Figure 5.

We further note that we do not include a k-means process in our PiCO method. PiCO is related to center-based clustering algorithms in our theoretical analysis. Since we restrict the gold label to be included in the candidate set, we believe that this piece of information could largely help avoid the bad optimum problem that occurs in a pure unsupervised setup. For the convergence properties of our PiCO algorithm, we did not empirically find any issues with PiCO converging to a (perhaps locally) optimal point. However, we want to refer the readers to the proof of k-means clustering in (Bottou & Bengio, 1994) for the convergence analysis.

**Discussion.** Regarding our empirical results where PiCO does indeed learn compact representations for each class, we can conclude that PiCO implicitly clusters data points in the contrastive embeddings space as a mixture of vMF distributions. In each iteration, our algorithm can be viewed as alternating the two steps until convergence, though different in detail. First, it is intractable to handle the whole training dataset, and thus we accelerate via a MoCo-style dictionary and MA-updated prototypes. Second, the contrastive loss also encourages the *uniformity* of the embeddings to maximally preserve information, which serves as a regularizer and typically leads to better representation. Finally, we use two copies of data examples such that data are also aligned in its local

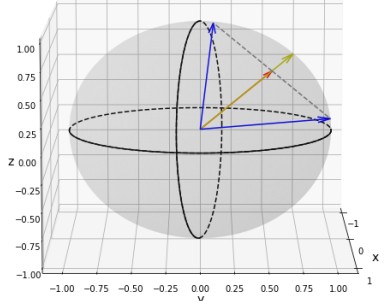 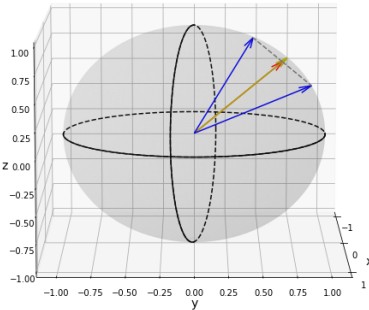

Figure 5: A large norm of Euclidean's mean vector also leads to a strong concentration of unit vectors to its mean direction.

region. Moreover, our theoretical result also answers why merely taking the pseudo-labels to select positive examples also leads to superior results, since the selected positive set will be refined as the training procedure proceeds.

Our theoretical results are also generalizable to other contrastive learning methods. For example, the classical unsupervised contrastive learning (He et al., 2020) actually assumes $n$-prototypes to cluster all locally augmented data; prototypical contrastive learning (Li et al., 2021b) directly assigns the data examples to one cluster to get the posterior, since there are no supervision signals; supervised contrastive learning (Khosla et al., 2020) chooses the known ground-truth as the posterior. In our problem, we follow two extreme settings to progressively obtain an accurate posterior estimation. Finally, it is also noteworthy that the objective in Eq. (9) has a close connection to the intraclass deviation (Saunshi et al., 2019), minimizing which is proven to be beneficial in obtaining tighter generalization error bound on downstream tasks. It should further be noted that our work differs from existing clustering-based CL methods (Caron et al., 2020; Li et al., 2021b), which explicitly involves clustering to aggregate the embeddings; instead, our results are derived from the loss itself.

# B  ADDITIONAL EXPERIMENTAL SETUPS AND RESULTS

## B.1  IMPLEMENTATION DETAILS

Following the standard experimental setup in PLL (Feng et al., 2020b; Wen et al., 2021), we split a clean validation set (10% of training data) from the training set to select the hyperparameters. After that, we transform the validation set back to its original PLL form and incorporate it into the training set to accomplish the final model training. We use an 18-layer ResNet as the backbone for feature extraction. Most of experimental setups for the contrastive network follow previous works (He et al., 2020; Khosla et al., 2020). The projection head of the contrastive network is a 2-layer MLP that outputs 128-dimensional embeddings. We use two data augmentation modules SimAugment (Khosla et al., 2020) and RandAugment (Cubuk et al., 2019) for query and key data augmentation respectively. Empirically, we find that even weak augmentation for key embeddings also leads to good results. The size of the queue that stores key embeddings is fixed to be 8192. The momentum coefficients are set as 0.999 for contrastive network updating and $\gamma = 0.99$ for prototype calculation. For pseudo target updating, we linearly ramp down $\phi$ from 0.95 to 0.8. The temperature parameter is set as $\tau = 0.07$. The loss weighting factor is set as $\lambda = 0.5$. The model is trained by a standard SGD optimizer with a momentum of 0.9 and the batch size is 256. We train the model for 800 epochs with cosine learning rate scheduling. We also empirically find that classifier warm-up leads to better performance when there are many candidates. Hence, we disable contrastive learning in the first 100 epoch for CIFAR-100 with $q = 0.1$ and 1 epoch for the remaining experiments.

## B.2  ABLATION STUDIES

**Moving-average updating factor $\phi$.**  We first present more ablation results about the effect of pseudo target updating factor $\phi$ on PiCO performance. Figure 6 (a) shows the results on two datasets

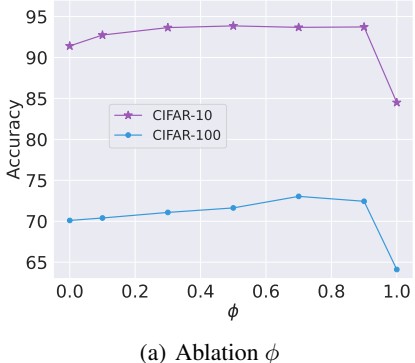 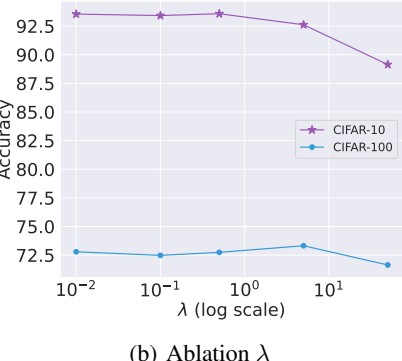

(a) Ablation $\phi$                    (b) Ablation $\lambda$

Figure 6: More ablation results on CIFAR-10 ($q = 0.5$) and CIFAR-100 ($q = 0.05$). (a) Performance of PiCO with varying $\phi$. (b) Performance of PiCO with varying $\lambda$.

CIFAR-10 ($q = 0.5$) and CIFAR-100 ($q = 0.05$). The overall trends on both datasets follow our arguments in section 4.3. Specifically, performance on CIFAR-100 achieves the best result when $\phi = 0.7$, and slight drops when $\phi = 0.9$. Therefore, in practice, we may achieve a better result with careful fine-tuning on $\phi$ value. In contrast, PiCO works well in a wide range of $\phi$ values on CIFAR-10. The reason might be that CIFAR-10 is a simpler version of CIFAR-100, and thus the prototypes can be high-quality quickly. But, setting $\phi$ to either 0 or 1 leads to a worse result, which has been discussed earlier.

**Loss weight $\lambda$.** Figure 6 (b) reports the performance of PiCO with varying $\lambda$ values that trade-off the classification and contrastive losses. $\lambda$ is selected from $\{0.01, 0.1, 0.5, 5, 50\}$. We can observe that on CIFAR-10, the performance is stable, but on CIFAR-100, the best performance is obtained at $\lambda = 5$. When $\lambda = 50$, PiCO shows inferior results on both two datasets. In general, a relatively small $\lambda$ ($< 10$) usually leads to good performance than a much larger value. When $\lambda$ is large, the contrastive network tends to fit noisy labels at the early stage of training.

Table 4: Performance of PiCO with varying $\gamma$ on CIFAR-10 and CIFAR-100.

| Dataset | $\gamma = 0.1$ | $\gamma = 0.5$ | $\gamma = 0.9$ | $\gamma = 0.99$ | $\gamma = 0.999$ |
|---|---|---|---|---|---|
| CIFAR-10 ($q = 0.5$) | 93.61 | 93.51 | 93.52 | 93.58 | 93.66 |
| CIFAR-100 ($q = 0.05$) | 72.87 | 73.09 | 72.54 | 72.74 | 67.33 |

**Prototype updating factor $\gamma$.** Finally, we show the effect of $\gamma$ that controls the speed of prototype updating and the results are listed in Table 4. On the CIFAR-10 dataset, the performance is stable with varying $\gamma$. But, on CIFAR-100, it can be seen that too large $\lambda$ leads to a significant performance drop, which may be caused by insufficient label disambiguation.

Table 5: Training accuracy of pseudo targets on CIFAR-10 and CIFAR-100.

| Dataset | CIFAR-10 | | | CIFAR-100 | | |
|---|---|---|---|---|---|---|
| | $q = 0.1$ | $q = 0.3$ | $q = 0.5$ | $q = 0.01$ | $q = 0.05$ | $q = 0.1$ |
| Accuracy | 98.28 | 98.26 | 96.79 | 99.06 | 96.27 | 90.58 |

**The disambiguation ability of PiCO.** Finally, we evaluate the disambiguation ability of the proposed PiCO. To see this, we calculate the max confidence to represent the uncertainty of an example, which has been widely used in recent works (Hendrycks & Gimpel, 2017). If one example is uncertain about its ground-truth, then it typically associates with low max confidence $\max_j s_j$. To represent the uncertainty of the whole training dataset, we calculate the mean max confidence (MMC) score. In Figure 7, we plot the MMC scores of different label disambiguation strategies in different training epochs. First, we can observe the MMC score of PiCO smoothly increases and finally achieves near 1 results, which means most of the labels are well disambiguated. In contrast,

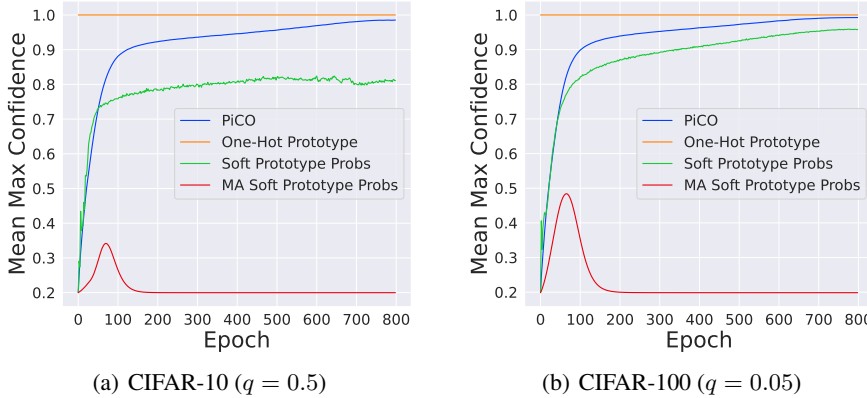

(a) CIFAR-10 ($q = 0.5$)    (b) CIFAR-100 ($q = 0.05$)

Figure 7: The mean max confidence curves of different label disambiguation strategies.

the Soft Prototype Probs strategy oscillates at the beginning, and then also increases to a certain value, which means that directly adopting the soft class probability also helps disambiguation. But, it is worth noting that it ends with a smaller MMC score compared with PiCO. The reason might be that the cosine distances to non-ground-truth prototypes are still at a scale. Hence, the Soft Prototype Probs strategy always holds a certain degree of ambiguity. Finally, we can see that the MA Soft Prototype Probs strategy fails to achieve great disambiguation ability. Compared with the non-moving-average version, it fails to get rid of severe label ambiguity and will finally converge to uniform pseudo targets again.

Furthermore, we evaluated the accuracy of the pseudo targets over training examples. From Table 5, we can find that the pseudo targets achieve high training accuracy. Combined with the fact that the mean max confidence score of the pseudo target is close to 1, the training examples finally become near supervised ones. Thus, the proposed PiCO is able to achieve near-supervised learning performance, especially when the label ambiguity is not too high. These results verify that PiCO has a strong label disambiguation ability to handle the PLL problem.

### B.3 ADDITIONAL RESULTS ON FINE-GRAINED CLASSIFICATION

In the sequel, we show the full setups and experimental results on fine-grained classification datasets. In particular, on CUB-200, we set the length of the momentum queue as $4192$. For CUB-200, we set the input image resolution as $224 \times 224$ and select $q \in \{0.01, 0.05, 0.1\}$. When $q = 0.05/0.1$, we warm up for $20/100$ epochs and train the model for $200/300$ epochs, respectively. For CIFAR-100-H, we select $q \in \{0.1, 0.5, 0.8\}$ and warm up the model for $100$ epochs when $q = 0.8$. Other hyperparameters are the same as our default setting. The baselines are also fine-tuned to achieve their best results.

From Table 6, we can observe that PiCO significantly outperforms all the baselines mentioned in section 4 on all the datasets. Moreover, as the size of candidate sets grows larger, PiCO consistently leads by an even wider margin. For example, on CIFAR-100-H, compared with the best baseline, performance improvement reaches **9.50%**, **11.15%** and **22.53%** in accuracy when $q = 0.1, 0.5, 0.8$, respectively. The comparison emerges the dominance of our label disambiguation strategy among semantically similar classes.

### B.4 STRATEGIES FOR POSITIVE SELECTION

While our positive set selection strategy is simple and effective, one may still explore more complicated strategies to boost performance. We have empirically tested two strategies: 1) **Filter-based**: we set a filter $\frac{|Y_i \cap Y_j|}{|Y_i \cup Y_j|} \leq \rho$ ($\rho = 0.5$) to remove example pairs who have dissimilar candidate sets at the early stage. 2) **Threshold-based**: we set a threshold $\max f^j(\mathrm{Aug}_q(\boldsymbol{x})) \leq \delta$ ($\delta = 0.95$) to remove those uncertain examples at the end of training, which has been widely used in semi-supervised learning (Sohn et al., 2020). Our basic principle is that contrastive learning is robust to noisy nega-

Table 6: Accuracy comparisons on fine-grained datasets. Bold indicates superior results.

| Dataset | Method | $q = 0.1$ | $q = 0.5$ | $q = 0.8$ |
|---|---|---|---|---|
| CIFAR-100-H | PiCO (ours) | **73.41** $\pm$ 0.27% | **72.04** $\pm$ 0.31% | **66.17** $\pm$ 0.23% |
| | LWS | 62.41 $\pm$ 0.03% | 57.25 $\pm$ 0.02% | 20.64 $\pm$ 0.48% |
| | PRODEN | 62.91 $\pm$ 0.01% | 60.89 $\pm$ 0.03% | 43.64 $\pm$ 1.82% |
| | CC | 50.40 $\pm$ 0.20% | 42.60 $\pm$ 0.11% | 37.80 $\pm$ 0.09% |
| | MSE | 46.05 $\pm$ 0.17% | 39.52 $\pm$ 0.28% | 15.18 $\pm$ 0.73% |
| | EXP | 45.73 $\pm$ 0.22% | 35.08 $\pm$ 1.71% | 22.31 $\pm$ 0.39% |

| Dataset | Method | $q = 0.01$ | $q = 0.05$ | $q = 0.1$ |
|---|---|---|---|---|
| CUB-200 | PiCO (ours) | **74.14** $\pm$ 0.24% | **72.17** $\pm$ 0.72% | **62.02** $\pm$ 1.16% |
| | LWS | 73.74 $\pm$ 0.23% | 39.74 $\pm$ 0.47% | 12.30 $\pm$ 0.77% |
| | PRODEN | 72.34 $\pm$ 0.04% | 62.56 $\pm$ 0.10% | 35.89 $\pm$ 0.05% |
| | CC | 56.63 $\pm$ 0.01% | 55.61 $\pm$ 0.02% | 17.01 $\pm$ 1.44% |
| | MSE | 61.12 $\pm$ 0.51% | 22.07 $\pm$ 2.36% | 11.40 $\pm$ 2.42% |
| | EXP | 55.62 $\pm$ 2.25% | 9.44 $\pm$ 2.32% | 7.3 $\pm$ 0.99% |
| | Fully Supervised | | 76.02 $\pm$ 0.19% | |

tive pairs and thus, we can flip those less reliable positive pairs to negative. Unfortunately, we did not observe statistically significant improvement to our vanilla strategy in experiments. These negative results suggest that the proposed PiCO has a strong error correction ability, which corroborates our theoretical analysis.

## B.5 THE INFLUENCE OF DATA GENERATION

In practice, some labels may be more analogous to the true label than others, which makes their probability of label flipping $q$ larger than others. In other words, the data generation procedure is non-uniform. In Section 4.4, we have shown one such case on CIFAR-100-H, where semantically similar labels have a larger probability of being a false positive. Moreover, we follow (Wen et al., 2021) to conduct empirical comparisons on data with alternative generation processes. In particular, we test two commonly used cases on CIFAR-10 with the following flipping matrix, respectively:

Table 7: Accuracy comparisons with different data generation processes on CIFAR-10.

| Method | Case (1) | Case (2) |
|---|---|---|
| PiCO | **94.49** $\pm$ 0.08% | **94.11** $\pm$ 0.25% |
| LWS | 90.78 $\pm$ 0.01% | 68.37 $\pm$ 0.04% |
| PRODEN | 90.53 $\pm$ 0.01% | 87.02 $\pm$ 0.02% |
| CC | 75.81 $\pm$ 0.13% | 66.51 $\pm$ 0.11% |
| MSE | 68.11 $\pm$ 0.23% | 39.49 $\pm$ 0.41% |
| EXP | 71.62 $\pm$ 0.79% | 48.87 $\pm$ 2.32% |

$$(1) = \begin{bmatrix} 1 & 0.5 & 0 & \cdots & 0 \\ 0 & 1 & 0.5 & \cdots & 0 \\ \vdots & & \cdots & & \vdots \\ 0.5 & 0 & 0 & \cdots & 1 \end{bmatrix}, \quad (2) = \begin{bmatrix} 1 & 0.9 & 0.7 & 0.5 & 0.3 & 0.1 & 0 & \cdots & 0 \\ 0 & 1 & 0.9 & 0.7 & 0.5 & 0.3 & 0.1 & \cdots & 0 \\ \vdots & & & & \cdots & & & & \vdots \\ 0.9 & 0.7 & 0.5 & 0.3 & 0.1 & 0 & 0 & \cdots & 1 \end{bmatrix}$$

where each entry denotes the probability of a label being a candidate. As shown in Table 7, PiCO outperforms other baselines in both cases. It is worth noting that in Case (2), each ground-truth label has a maximum probability of 0.9 of being coupled with the same false positive label. In such a challenging setup, PiCO still achieves promising results that are competitive with the supervised performance, which further verifies its strong disambiguation ability.

## B.6 THE INFLUENCE OF PROTOTYPE CALCULATION

There are several ways to calculate the prototypes and hence, we further test a variant of PiCO that re-computes the prototypes by averaging embeddings of all training examples at the end of each epoch. We train the models using one Quadro P5000 GPU respectively and evaluate the average training time per epoch. From Table 8, we can observe that the Re-Compute variant achieves competitive results, but is much slower than PiCO.

Table 8: Training time (min/epoch) and accuracy of different prototype calculation methods.

| Dataset | Method | Time | Accuracy |
|---|---|---|---|
| CIFAR-10 ($q = 0.5$) | PiCO | 0.94 | 93.58 |
| | Re-Compute | 1.39 | 93.55 |
| CIFAR-100 ($q = 0.05$) | PiCO | 0.96 | 72.74 |
| | Re-Compute | 1.40 | 72.35 |

---

**Algorithm 1:** Pseudo-code of PiCO (one epoch).

---

1 **Input:** Training dataset $\mathcal{D}$, classifier $f$, query network $g$, key network $g'$, momentum queue, uniform pseudo-labels $\boldsymbol{s}_i$ associated with $\boldsymbol{x}_i$ in $\mathcal{D}$, class prototypes $\boldsymbol{\mu}_j$ ($1 \leq j \leq C$).

2 **for** $iter = 1, 2, \ldots,$ **do**

3     sample a mini-batch $B$ from $\mathcal{D}$

    // query and key embeddings generation

4     $B_q = \{\boldsymbol{q}_i = g(\text{Aug}_q(\boldsymbol{x}_i)) | \boldsymbol{x}_i \in B\}$

5     $B_k = \{\boldsymbol{k}_i = g'(\text{Aug}_k(\boldsymbol{x}_i)) | \boldsymbol{x}_i \in B\}$

6     $A = B_q \cup B_k \cup \text{queue}$

7     **for** $\boldsymbol{x}_i \in B$ **do**

        // classifier prediction

8         $\tilde{y}_i = \arg\max_{j \in Y_i} f^j(\text{Aug}_q(\boldsymbol{x}_i))$

        // momentum prototype updating

9         $\boldsymbol{\mu}_c = \text{Normalize}(\gamma\boldsymbol{\mu}_c + (1 - \gamma)\boldsymbol{q}_i)$, if $\tilde{y}_i = c$

        // positive set generation

10        $P(\boldsymbol{x}_i) = \{\boldsymbol{k}' | \boldsymbol{k}' \in A(\boldsymbol{x}_i), \tilde{y}' = \tilde{y}_i\}$

11     **end**

    // prototype-based label disambiguation

12     **for** $\boldsymbol{q}_i \in B_q$ **do**

13         $\boldsymbol{z}_i = \text{OneHot}(\arg\max_{j \in Y_i} \boldsymbol{q}_i^\top \boldsymbol{\mu}_j)$

14         $\boldsymbol{s}_i = \phi\boldsymbol{s}_i + (1 - \phi)\boldsymbol{z}_i$

15     **end**

    // contrastive loss calculation

16     $\mathcal{L}_{\text{cont}}(g; \tau, A) = \frac{1}{|B_q|} \sum_{\boldsymbol{q}_i \in B_q} \left\{ -\frac{1}{|P(\boldsymbol{x}_i)|} \sum_{\boldsymbol{k}_+ \in P(\boldsymbol{x}_i)} \log \frac{\exp(\boldsymbol{q}_i^\top \boldsymbol{k}_+/\tau)}{\sum_{\boldsymbol{k}' \in A(\boldsymbol{x}_i)} \exp(\boldsymbol{q}_i^\top \boldsymbol{k}'/\tau)} \right\}$

    // classification loss calculation

17     $\mathcal{L}_{\text{cls}}(f; B) = \frac{1}{|B|} \sum_{\boldsymbol{x}_i \in B} \sum_{j=1}^{C} -s_{i,j} \log(f^j(\text{Aug}_q(\boldsymbol{x}_i)))$

    // network updating

18     minimize loss $\mathcal{L} = \mathcal{L}_{\text{cls}} + \lambda\mathcal{L}_{\text{cont}}$

    // update the key network and momentum queue

19     momentum update $g'$ by using $g$

20     enqueue $B_k$ and classifier predictions and dequeue

21 **end**

---

## C   PSEUDO-CODE OF PICO

We summarize the pseudo-code of our PiCO method in Algorithm 1.

## D   THE LITERATURE OF PROTOTYPE LEARNING

Prototype learning (PL) aims to learn a metric space where examples are enforced to be closer to its class prototype. PL is typically more robust in handling few-shot learning (Snell et al., 2017), zero-shot learning (Xu et al., 2020), and out-of-distribution samples (Arik & Pfister, 2019). Recently, PL has demonstrated promising results in weakly-supervised learning, such as semi-supervised learning (Han et al., 2020), noisy-label learning (Li et al., 2021a), etc. For example, USADTM (Han et al., 2020) shows that informative class prototypes usually lead to better pseudo-labels for semi-supervised learning than classical pseudo-labeling algorithms (Sohn et al., 2020) which reuse classifier outputs. Motivated by this, we also employ contrastive prototypes for label disambiguation.

