# OpenReview forum: "PiCO: Contrastive Label Disambiguation for Partial Label Learning"
_ICLR.cc/2022/Conference — ICLR 2022 Oral_

### Official Review · Reviewer_fAwc · 2021-10-26

**Correctness:** 3
**Technical Novelty And Significance:** 3
**Empirical Novelty And Significance:** 3
**Recommendation:** 8
**Confidence:** 4

**Main Review:**

I have the following comments.
Pros:
1.	The structure of this paper is well-written and easy to follow. The motivation is clear, and the solution is simple but effective.
2.	I think the theoretical justification of the proposed method from the expectation-maximization perspective is very interesting. It is a generic result and potentially help the community understand the property of contrastive learning.
3.	The experimental results are quite strong and the ablation study also looks good. I especially appreciate the results on new fine-grained datasets to test the performance of PLL methods.
Cons:
1.	Experiments are mainly conducted on datasets with uniformly generated candidates, it is not clear how PiCO performs on non-uniform datasets, such as in the label-dependent setting where the probability of label flipping depends on its ground-truth label.
2.	Why does the second equality in Eq. (9) hold? It would be appreciated if a clear explanation is provided.
3.	There is a redundant space notation in Table 1 (the method CC with q=0.5 on CIFAR10).


**Summary Of The Paper:**

This work approaches the partial label learning problem where each training example is annotated with multiple candidate labels, in contrast to the conventional supervised learning setup that the ground-truth label is provided. The proposed framework comprises two key components: (1) a contrastive learning module that uses the classifier output to select positive pairs and (2) a label disambiguation method that uses the contrastive prototypes to update the pseudo targets in a moving-average style. The experimental results look quite strong, where the performance of PiCO nearly approaches the fully-supervised results.

**Summary Of The Review:**

This paper is clear and technically solid. The theoretical justification is interesting.

---

> ### Author Response · Authors · 2021-11-14
> **Response to Reviewer fAwc**
>
> Thank you! As a briefing, we added the following section in our revision to respond to the reviews: (I) we added experiments in the label-dependent setup in Appendix B.5; (ii) we added derivation of the second equality in Eq. (9) in Appendix A. We provide a more detailed explanation and responses as the following.
>
> **Q1: Experiments are mainly conducted on datasets with uniformly generated candidates, it is not clear how PiCO performs on non-uniform datasets, such as in the label-dependent setting where the probability of label flipping depends on its ground-truth label.**
>
> **A:** This is indeed a good point, thank you! In our original version, we provide an experimental validation on the non-uniform case for CIFAR-100 with hierarchical classes, where false positive labels are generated within the same superclass.
>
> Additionally, we add a new set of experiments for the label-dependent setting shown in Appendix B.5. With slightly more details, we closely follow [1] and generate false candidates with varying probabilities w.r.t. the ground-truth. The results demonstrate that PiCO achieves significantly better performance than the baselines under this setup.
>
> [1] Hongwei Wen, Jingyi Cui, Hanyuan Hang, Jiabin Liu, Yisen Wang, and Zhouchen Lin. Leveraged weighted loss for partial label learning. In ICML, volume 139 of Proceedings of Machine Learning Research, pp. 11091–11100. PMLR, 2021.
>
> **Q2: Why does the second equality in Eq. (9) hold? It would be appreciated if a clear explanation is provided.**
>
> **A:** We are sorry for the lack of clearness. We have added more clarification and derivation in the revision. We may refer the reviewer to Appendix A in the revised manuscript. Thank you for pointing this out.
>
> **Q3: There is a redundant space notation in Table 1 (the method CC with q=0.5 on CIFAR10).**
>
> **A:** Our bad! Please check the revision.

---

### Official Review · Reviewer_k7zr · 2021-11-01

**Correctness:** 4
**Technical Novelty And Significance:** 4
**Empirical Novelty And Significance:** 4
**Recommendation:** 8
**Confidence:** 3

**Main Review:**

## Strengths

The problem that the paper tackles, namely partial label learning, is important in the real life where labelling is difficult due to semantic ambiguity (e.g. husky vs malamute). The problem has several connections especially with weakly supervised learning.

The approach proposed in the paper is very well motivated. The idea of making contrastive learning and prototype learning is sound as they can work collaboratively in the EM fashion (proved in the paper).

The approach is backed by very impressive empirical results. The presented analyses support the motivation of the approach, e.g. cluster visualisation shows that the clusters are well formed with few classification errors.

The paper is well written with a clear structure. Reproduction doesn't seem difficult.


## Weaknesses

As the approach is EM-like, it also inherits some cons from EM. This is however not presented in the paper. For instance, in which case the learning will get stuck in a bad optimum? What is the consequence? Is there any way to avoid that?

It is unclear how the models were fine-tuned. Did the authors use clean dev sets which don't have any false positive labels? If this is the case, would it be more realistic to fine-tune the models on noisy dev sets instead? One concern is that clean dev sets can provide lots of information for removing labelling ambiguity. For instance, one can achieve reasonably good prototype representations just from a few clean labelled instances.

It seems that the paper mentions too little about prototype learning literature.



**Summary Of The Paper:**

The paper proposes an innovative approach to partial label learning, where an instance is assigned with some false positive labels besides its true label (due to difficulty in labelling). The approach blends contrastive learning with prototype learning: (i) the former helps to form good clusters for the latter to learn prototype representations (ii) in return the latter helps to select positive samples for the former. Theoretically, the authors prove that the two of them work collaboratively in the EM fashion.

The paper demonstrates the effectiveness of the proposed approach using a typical setting for the task. Specifically, CIFAR 10 and CIFAR 100 were used and each instance was randomly assigned false labels with some probabilities. The proposed approach achieved impressive results, substantially outperforming five recent models in the literature. Moreover, its performance strongly approaches the one of supervised learning.

The paper also presents a much stricter setting when false positive labels are semantically correlated with true labels. The proposed approach also gained impressive results on CUB-200 and CIFAR-100-H datasets. Moreover, the paper shows several in-depth analyses.



**Summary Of The Review:**

This is a strong paper tackling an important problem. The approach is interesting and the technique is sound. The claims are well backed by impressive empirical result and theory. Therefore, overall, the contributions of the paper will have a certain impact to the community.

---

> ### Author Response · Authors · 2021-11-14
> **Response to Reviewer k7zr**
>
> Thank you very much for your comments and suggestions! We are happy you enjoyed the paper. We have updated our paper based on the suggestions and comments of the reviewers. Summary of revision: (i) we added the procedure of hyperparameter selection in Appendix B.1; (ii) we added related works about prototype learning literature in Appendix D.
>
> **Q1: As the approach is EM-like, it also inherits some cons from EM. This is however not presented in the paper. For instance, in which case the learning will get stuck in a bad optimum? What is the consequence? Is there any way to avoid that?**
>
> **A:** We acknowledge the bad optima problem with the conventional EM family algorithms. However, the PLL problem setup naturally differs because, as we describe in Section 2, it restricts the gold label to be included in the candidate set. In that regard, we believe that this piece of information could largely help avoid the bad optimum problem that occurs in a pure unsupervised setup --- for instance, running EM for unsupervised k-means clustering.
>
> In our extensive experiments on all well-known benchmarks in this field, we did not find this problem of falling into bad optima for PiCO. However, we do acknowledge that PiCO, or any other PLL methods, may have this problem when the supervision signal is too scarce. Technically, this may be alleviated by using standard techniques in the literature of the EM algorithm like [1]. Yet this may have gone beyond the scope of this paper where our focus is on a commonly adopted PLL setup.
>
> [1] Shaban A, Farajtabar M, Xie B, et al. Learning Latent Variable Models by Improving Spectral Solutions with Exterior Point Method[C]//UAI. 2015: 792-801.
>
> **Q2: It is unclear how the models were fine-tuned. Clean dev sets can provide lots of information for removing labeling ambiguity.**
>
> **A:** Generally, we strictly follow the standard procedure in the recent literature of PLL [2,3]. Namely, to conduct the hyperparameter selection, we rely on a clean dev set that is drawn from the training set at a ratio of 10%. After this step, we transform the clean dev set back to its original PLL form and incorporate it into the training set to accomplish the final model training.
> Alongside making the comparison with the prior works fair, we additionally want to refer the reviewer to Appendix B.2 for ablation results. PiCO does not seem to be sensitive to the hyperparameter choices. We hope this may mitigate the concern raised by the reviewer on the functionality of the clean dev set.
>
> [2] Lei Feng, Jiaqi Lv, Bo Han, Miao Xu, Gang Niu, Xin Geng, Bo An, and Masashi Sugiyama. Provably consistent partial-label learning. In NeurIPS, 2020.
>
> [3] Hongwei Wen, Jingyi Cui, Hanyuan Hang, Jiabin Liu, Yisen Wang, and Zhouchen Lin. Leveraged weighted loss for partial label learning. In ICML, volume 139 of Proceedings of Machine Learning Research, pp. 11091–11100. PMLR, 2021.
>
> **Q3: It seems that the paper mentions too little about prototype learning literature.**
>
> **A:** We have added more discussions about prototype learning in Appendix D.

---

> > ### Comment · Reviewer_k7zr · 2021-11-22
> > **thanks for the response**
> >
> > I would like to thank the authors for the informative response. Although I disagree with the setup using clean dev sets, I understand that this is common in several research areas (and sometimes also in unsupervised learning e.g. unsupervised parsing in NLP).
> >
> > Satisfied with the response, and the latest draft, I would like to keep my original score.

---

### Official Review · Reviewer_7qAy · 2021-11-03

**Correctness:** 3
**Technical Novelty And Significance:** 3
**Empirical Novelty And Significance:** 4
**Recommendation:** 8
**Confidence:** 3

**Main Review:**

Strengths
- Incredibly strong PLL performance close to supervised learning and substantially stronger than baselines.
- Helpful ablations and results on various datasets.
- Provides theoretical connection of the proposed method to EM.

Weaknesses
- More should be said about the closeness of PLL performance and supervised learning performance. This comment may seem counterintuitive, but I do not think this closeness can be so easily glossed over, especially when it is one of the main results.
- (minor) Treatment of clustering is simplistic. K-means is a very specific type of clustering algorithm, and only one of multiple classic clustering techniques.

Questions and comments:

- How is the “novel” prototype-based label disambiguation so different from using a softmax layer? A softmax layer will also have a prototype vector for each class.
- I can see the convenience of momentum style updating of prototype vectors, but re-computing them every iteration (or every N iterations) is not so expensive and at most increases training by 2x.


**Summary Of The Paper:**

The authors present a new technique for partial label learning (PLL). PLL is the task where the labels for each instance include both the ground truth label and a randomly sampled set of distractor labels, and during training the model learns a latent decision for which among this set is the ground truth. The technique presented by the authors uses a combination of momentum (in the representation) and contrastive learning (to augment the label set) that leads to improved PLL results, reaching nearly fully supervised performance.

**Summary Of The Review:**

This seems like a particularly strong result. More could be said about the implications of their results, but otherwise seems like a good paper.

---

> ### Author Response · Authors · 2021-11-14
> **Response to Reviewer 7qAy**
>
> Thanks very much for your insightful comments and suggestions! We have updated our paper based on the comments of the reviewers. Summary of revision: (i) we added discussion about the closeness of PLL performance and supervised learning performance in Appendix B.2; (ii) we tested the training time of different prototype calculation methods in Appendix B.6.
>
> **Q1：More should be said about the closeness of PLL performance and supervised learning performance. This comment may seem counterintuitive, but I do not think this closeness can be so easily glossed over, especially when it is one of the main results.**
>
> **A:** This is a very good point, thank you! Indeed, while we show that PiCO displays on-par results with a fully-supervised setup in Table 1, we want to identify that this closeness does depend on the dataset setup, in particular on the average number of false candidates $q$ in section 4.1. We show the most commonly used setup for $q$ in order to compare with prior works (Table 1). However, if we increase $q$, as the label space uncertainty gets aggravated, PiCO together with all other PLL methods would anticipate a performance drop.
>
> To further analyze this point, we added another set of empirical evaluations in our newest revision. We refer the reviewer to Table 5 in Appendix B.2, where we evaluate the training accuracy of the generated pseudo targets. As an example, we found that under the setting of $q=0.05$, the pseudo targets achieve a 96.27% training accuracy on CIFAR-100. In addition, in Figure 7, we show that the mean max confidence score of the pseudo target is close to 1 over the training dataset. This is somewhat a promising result to us because it demonstrates PiCO has managed to disambiguate a majority of the uncertain partial label sets. Not only does it interpret the performance closeness of PiCO against the fully-supervised setup, but also highlights the effectiveness of our contrastively trained representations in enabling strong label disambiguation.
>
> **Q2: Treatment of clustering is simplistic. K-means is a very specific type of clustering algorithm, and only one of multiple classic clustering techniques.**
>
> **A:** We apologize for this confusion! We do not include a K-means process in our PiCO method, as shown in the pseudo-code provided in Appendix C. PiCO is related to center-based clustering (e.g. K-means) in our theoretical analysis. We draw a connection between our contrastively learned representation in PiCO with the center-based clustering algorithms in Section 5. We will make this clearer in our revision.
>
> **Q3: How is the 'novel' prototype-based label disambiguation so different from using a softmax layer? A softmax layer will also have a prototype vector for each class.**
>
> **A:** Notably, the prototype vector calculated in PiCO is based on the contrastive learned embeddings. We may refer the reviewer to Eq. (8). The major difference between our prototypes and the softmax layer is attributed to the uniformity term (b) in Eq. (8). We argue that the minimization of such a term encourages information preservation in the contrastive embedding space, as proved in [1], which also makes the contrastive prototypes more representative.
>
> [1] Tongzhou Wang and Phillip Isola. Understanding contrastive representation learning through alignment and uniformity on the hypersphere. In ICML, volume 119 of Proceedings of Machine Learning Research, pp. 9929–9939. PMLR, 2020.
>
> **Q4: I can see the convenience of momentum style updating of prototype vectors, but re-computing them every iteration (or every N iterations) is not so expensive and at most increases training by 2x.**
>
> **A:** The reviewer rightfully pointed out a possible variant for the calculation of the prototypes. While re-computing the prototype over the entire training set per-iteration can be overly expensive, we add a new experiment where the prototypes are re-computed by averaging embeddings of all training examples at the end of each epoch. As we can see from Table 8, this variant does achieve the on-par result with PiCO but ends up with a much slower training speed.

---

> > ### Comment · Reviewer_7qAy · 2021-11-14
> > **quick points**
> >
> > Thanks for the nice work and detailed response. I wanted to quickly follow up on a couple points.
> >
> > 1. More should be said about the closeness of PLL performance and supervised learning performance.
> >
> > Sorry if I was not clear. I was hoping you would explain more why your performance in the paper is surprising or not. Certainly it is surprising to me that performance in PLL can be so close to fully supervised setting. This explanation does not necessarily need new empirical results, but some scientific hypothesis or additional analysis would be helpful. Perhaps I am missing something obvious and you can explain to me in the comments why the performance is surprising or not.
> >
> > I appreciate the connection to EM and feel it is related, as well are the results on fine-grained classification (table 3). For the sake of discussion, I also think that unsupervised clustering with image features may get close to supervised performance in some cases after mapping clusters with to labels, although I haven't recently looked at results for the relevant datasets.
> >
> > Also, I would be curious to know if you think the "closeness" is because supervised models still needs to be improved.
> >
> > 2. (new point) Are random labels chosen at beginning and kept throughout?
> >
> > This is perhaps a silly question, but I would like to double check anyway. I was looking at code provided in the supplementary material. It is not 100% clear to me that random "pseudo labels" are chosen at beginning and kept until the end of training. Can you please verify if this is the case? Similarly, I noticed that there is support for distributed training --- can you verify that all nodes are using the same set of randomly selected "pseudo labels"?
> >
> > ---
> >
> > typo: Loss wight

---

> > > ### Author Response · Authors · 2021-11-16
> > > **Response to Reviewer 7qAy for quick points**
> > >
> > > We refresh the answer down below. Hope this time the explanation could further help! Let us know if you have any further questions.
> > >
> > > **Q1. I was hoping you would explain more why your performance in the paper is surprising or not. Some scientific hypothesis or additional analysis would be helpful.**
> > >
> > > **A:** Got you! Basically, this performance closeness can be intuitively explained as follows. When the label disambiguation is perfectly accomplished, i.e. the label space uncertainty is reduced to zero, the PLL setup is equivalent to being fully supervised. Notice that this is also due to the fundamental assumption of the PLL --- the gold label appears in the label candidate set. From this standpoint, as we are discussing PiCO's excellent performances, it implies that PiCO achieves supreme label disambiguation effect thanks to the contrastive learned embeddings and the prototypes.
> > >
> > > Indeed, to comprehensively answer this question is very much multidisciplinary. It depends on the development of contrastive learning, PLL methods and etc. Through PiCO, we pioneer to validate the strong functionality of contrastively trained embeddings for PLL setup both empirically and theoretically. Though a bit unrelated, there has also been a few papers centered on the theory of pure contrastive learning (not for PLL) as in [1,2,3] that could be helpful to check. To sum up, we look forward to keeping the efforts pushing this direction of contrastive learning for PLL in the future.
> > >
> > > [1] Saunshi N, Plevrakis O, Arora S, et al. A theoretical analysis of contrastive unsupervised representation learning[C]//International Conference on Machine Learning. PMLR, 2019: 5628-5637.
> > >
> > > [2] HaoChen J Z, Wei C, Gaidon A, et al. Provable Guarantees for Self-Supervised Deep Learning with Spectral Contrastive Loss[J]. arXiv preprint arXiv:2106.04156, 2021.
> > >
> > > [3] Zhang Y, Hooi B, Hu D, et al. Unleashing the Power of Contrastive Self-Supervised Visual Models via Contrast-Regularized Fine-Tuning[J]. arXiv preprint arXiv:2102.06605, 2021.
> > >
> > > **Q2. For the sake of discussion, I also think that unsupervised clustering with image features may get close to supervised performance in some cases after mapping clusters with to labels, although I haven't recently looked at results for the relevant datasets.**
> > >
> > > **A:** The answer to this question very much overlaps with Q1 :). The unsupervised clustering perhaps also has room to improve, indicated by the performance of PiCO on PLL benchmarks. However, there is one phenomenon difference between unsupervised setting and the PLL is with the acquirement of the gold label in the candidate set serving as a weak supervision signal. Further, whether the unsupervised clustering could get close or not to the supervised performance also depends on other factors, such as the complexity of the data distribution and etc. While think this is definitely an interesting direction, we do posit this goes way beyond the scope of this paper --- with PiCO, we are dedicated to the PLL setting and standardized benchmarks.
> > >
> > > **Q3. Also, I would be curious to know if you think the "closeness" is because supervised models still needs to be improved.**
> > >
> > > **A:** As well, the answer to this question can also be related to Q1. We cannot be sure if the supervised models still need to be improved, but as we mentioned above, in the optimal condition where the label disambiguation is perfectly done, the PLL is equivalent to a supervised setup.
> > >
> > > **Q4. Are random labels chosen at beginning and kept throughout? It is not 100% clear to me that random "pseudo labels" are chosen at beginning and kept until the end of training.**
> > >
> > > **A:** If we get the reviewer right, the ''pseudo labels'' here may refer to the ''pseudo targets''. And the answer to this question is that the pseudo targets are set as uniform vectors $s_j=\frac{1}{|Y|}\mathbb{I}(j\in Y)$ (not random) at the beginning and updated by using the candidate label sets as well as the prototypes. In addition, the true gold labels were never revealed to the model, nor any part involved in the training.
> > >
> > > The attribute ```confidence``` in Python class ```Partial_loss``` stands for the pseudo targets. We refer the reviewer to ```utils/utils_loss.py, line 5-29``` in our codes.
> > >
> > > **Q5. Can you verify that all nodes are using the same set of randomly selected "pseudo labels"?**
> > >
> > > **A:** This is certainly our bad in terms of writing. Technically, we implemented PiCO using a distributed setup because we thought parallelization would be needed. However, while we train PiCO we had never actually enabled the distributed setup but only resorted to one single GPU training. We will definitely make this clearer in the code that we release upon publication. Hope this helps!
> > >
> > > **Q6. typo: Loss wight**
> > >
> > > **A:** Thank you!

---

> > > > ### Comment · Reviewer_7qAy · 2021-11-16
> > > > **re: supervised comparison**
> > > >
> > > > > When the label disambiguation is perfectly accomplished, i.e. the label space uncertainty is reduced to zero, the PLL setup is equivalent to being fully supervised.
> > > >
> > > > I see. This is roughly backed up by the strong training accuracy performance in Table 5.
> > > >
> > > > In general, I am not sure that optimizing your loss function guarantees for the "gold label" to be predicted, since there is often natural noise in the data due to natural ambiguity and/or human error in annotation, but I am not sure to what extent this is for the relevant datasets.
> > > >
> > > > Again, sorry if I was not being clear and let me be more direct. Although I gave the paper a high score, I still have some skepticism of the results because the performance for PLL is so close to supervised learning. I have some small concerns that either a) somehow the model is cheating during training --- one way this could happen is if the pseudo targets are randomly chosen every epoch, since the supervision signal from the gold label would be stronger as it would always be included, or b) the supervised learning baseline is a weak baseline.
> > > >
> > > > > supervised models
> > > >
> > > > After further inspection, it appears supervised models typically perform better than the numbers included in this paper. For example, Wide ResNet (2016) scores about 96% accuracy on CIFAR-10 and 81% accuracy on CIFAR-100.
> > > >
> > > > For CUB-200 I assume you are using the configuration where bounding box is not provided at train or test time. I think that supervised Bilinear-CNN (ICCV 2015) achieves higher than 80% accuracy. Is there a reason why you did not provided supervised results for CUB-200, and do you have ideas why the performance gap is bigger for CUB-200 than CIFAR-10 or CIFAR-100?
> > > >
> > > > Perhaps I have misunderstood whether these supervised baselines are directly comparable with your results, so feel free to clear things up. :)

---

> > > > > ### Author Response · Authors · 2021-11-17
> > > > > **Response to Reviewer 7qAy for supervised comparison**
> > > > >
> > > > > Thanks for keeping up the discussion.
> > > > >
> > > > > First, we are absolutely sure about our code and method implementation. We fully guarantee the correctness and reproducibility of the code. That said, with numerous rounds of code reviewing, we do assert to the reviewer that the correctness of our code and the cases the review had pointed out is not presented in PiCO. Also, we uploaded the code at the time of submission, please feel free to check/run it, by running ```bash run.sh```.
> > > > >
> > > > > Second, we think you are absolutely right about the natural noise appearing in the dataset. We do *not* claim PiCO, nor any other approaches proposed in the regime of PLL, can get close to the supervised performance in any case. As we simply adopt the exact identical experimental protocol and the dataset, the outcome of this paper is that PiCO is able to approach the supervised performance under certain scenarios and on certain datasets. We do extrapolate that when the natural noise is more widely possessed, the partial label setting might be more challenging especially compared to a full-supervised setup.
> > > > >
> > > > > Third, we respectfully disagree with the reviewer that our baseline is not strong enough. For one thing, we did carefully tune our ResNet-18-based baseline model. The others results can be better like the reviewer rightfully listed out; nonetheless, with a stronger backbone model such as WideResNet and PreActResNet, the resulting enhancement is not surprising. We chose the original ResNet18 architecture as it is one of the most commonly used in the AI community. Further, this paper is very much not about architectural choices but is more emphatical about the label disambiguation algorithm and contrastive learning. For another, we may reference some high-star open-source GitHub repositories, such as https://github.com/kuangliu/pytorch-cifar, where our ResNet-18-backbone baseline yields a lower **93.02%** accuracy on CIFAR-10.
> > > > >
> > > > > Last but not least, we kindly ask the reviewer to consider PiCO from a label disambiguation and partial label learning standpoint. As we mentioned, the PLL setup is conceptually reduced to a full-supervised setup and the core challenge is just how to perform the label disambiguation. Put it another way, the excellent performance from PiCO gives us very strong confidence in the capacity of this framework on label disambiguation. In the regime of partial label learning, this is, in our honest opinions, a very promising result/framework that is worthy of spreading.
> > > > >
> > > > > In what follows, we also provide detailed clarification for addressing the concerns of the Reviewer.
> > > > >
> > > > > **Q1. somehow the model is cheating during training --- one way this could happen is if the pseudo targets are randomly chosen every epoch since the supervision signal from the gold label would be stronger as it would always be included.**
> > > > >
> > > > > **A:** Sorry for confusing the reviewer. We suppose here "pseudo targets" refers to the "candidate label set", which defines the range that the pseudo targets being updated. It can be checked in our codes that the candidate label sets are generated at the beginning and we **never change** the value of the candidate label sets anymore. When training, the ```DataLoader (train_loader)``` samples data examples with the originally generated candidate label sets. **Nothing will be randomly reset during training.** We refer the reviewer for the attribute ```given_label_matrix``` (i.e. the candidate set) of class ```CIFAR10_Augmentention``` in ```utils/cifar10.py``` and ```utils/cifar100.py``` to see whether it is modified or randomly reset during training.
> > > > >
> > > > > **Q2: The supervised learning baseline is a weak baseline.**
> > > > >
> > > > > **A:** We strictly implemented the supervised learning model by using the same experimental setup as PiCO. We used ResNet18 and contrastive learning, which definitely leads to different results. But, we believe we have reported **fair** results for supervised learning models. We also note that we have provided demo codes as well as a shell file ```run.sh``` for running the code, the reviewer can set the parameter ```--partial_rate 0.0``` to train a supervised model and check whether our results are real or not.
> > > > >
> > > > > **Q3: Is there a reason why you did not provided supervised results for CUB-200, and do you have ideas why the performance gap is bigger for CUB-200 than CIFAR-10 or CIFAR-100?**
> > > > >
> > > > > **A:** That's definitely our bad. The answer is the same as Q2 that we used different experimental setups but the results are fair for all the comparing methods. We will report the supervised results of CUB-200 soon.

---

> > > > > > ### Comment · Reviewer_7qAy · 2021-11-22
> > > > > > **final comments**
> > > > > >
> > > > > > Thank you for the detailed responses. I feel more confident in this work, especially after you provided your explanation and new results. I will keep my original score of 8.

---

> > > > > > > ### Author Response · Authors · 2021-11-22
> > > > > > > **Thanks for discussion!**
> > > > > > >
> > > > > > > We thank the reviewer for the extra time and effort spent. We definitely believe that the discussion in the last few days can make our paper more solid. We will incorporate the new results and the fruitful points in our future revision.

---

> > > > > ### Author Response · Authors · 2021-11-18
> > > > > **Added supervised results on CUB-200**
> > > > >
> > > > > Dear reviewer, as you may have requested, we just uploaded another revision adding the supervised results on CUB-200 dataset. In Table 6 of Appendix B.3, we can see that PiCO achieves competitive results compared to the supervised setup ($-1.88$% Accuracy) when $q=0.01$. We can also see that when $q=0.1$,   while still with the best results among the competitors, the performance discrepancy between PiCO and the supervised results is enlarged to $13.00$% inaccuracy. These empirical findings are still consistent with our former responses. Hope this is helpful!

---

### Decision · Program_Chairs · 2022-01-20

**Decision:**

Accept (Oral)

**Comment:**

This paper presents PiCO, a novel approach for partial label learning, which achieves very strong performance close to that of fully supervised learning and outperforms PPL baselines. The experiments are extensive with very impressive results and the analysis are thorough.